# Combining Pretrained Tabular Models with Static GNNs in Relational Deep Learning

## Abstract

Relational databases, organized into tables connected by primary-foreign key relationships, are widely used in industry. Companies leverage this data to build highly accurate, feature-engineered tabular models—often using boosted decision trees—to predict key metrics such as customer transactions and product revenues. However, these models need frequent retraining as new data is introduced, which is both expensive and time-consuming. Despite this, by being the result of extensive engineering effort, they remain difficult to outperform using generalist methods, like Temporal Graph Neural Networks (TGNNs) trained over the same relational data. Rather than attempting to replace tabular models with generalist approaches, we propose to combine the strengths of tabular models and static Graph Neural Networks (GNNs). GNNs offer better speed and scalability than TGNNs, and, as we argue, the primary strength of graph representation learning for these tasks does not lie in modeling temporal dynamics—something highly-engineered tabular models excel at—but in capturing complex relationships within the database, which are hard to featurize. Our approach integrates *all* predictive embeddings of *all* tabular models developed for various tasks into a single static GNN framework. Experimental results on the RelBench benchmark show that our approach achieves a performance improvement of up to 33% and an inference speedup of up to 1050x, making it highly suitable for real-time inference.

## 1 Introduction

Relational databases are extensively used in industry due to their flexibility, extensibility, and speed (Wheeler et al., 2000; Kremer, 2006; Johnson et al., 2016). The information, organized into tables, not only record entities, their features, and their relations (via primary and foreign key relationships), but also records events, such as transactions, with their associated timestamps. The tabular format simplifies data maintenance and enhances data accessibility and retrieval through query languages like SQL (Codd, 1970; Chamberlin & Boyce, 1974). Due to the presence of both timestepped events and relations, tasks over relational databases tend to be both temporal and relational, such as forecasting future product sales and predicting future customer purchases and churn (Robinson et al., 2024). For decades, companies have been building in-house predictive models over relational databases by creating meticulously engineered relational and temporal features (Dong & Liu, 2018; Ganguli & Thakur, 2020) that flatten the complex temporal-relational data into a single table, which is then used as input for tabular models like XGBoost and LightGBM (Ke et al., 2017; Chen & Guestrin, 2016). However, incorporating new data sources or addressing new tasks with these tabular models is both costly and time-consuming (Heaton, 2016).

Generalist models such as Temporal Graph Neural Networks (TGNNs) (Rossi et al., 2020; Dileo et al., 2023; Cini et al., 2023; Longa et al., 2023) promise to offer a more cost-and-time effective alternative to tabular models (Fey et al., 2024), and have shown some early success (Robinson et al., 2024). These generalist approaches are applicable as relational databases can be naturally represented as temporal, heterogeneous graphs, where each row in a table corresponds to an attributed node, and the edges are defined by primary-foreign key relationships. However, generalist models often struggle to outperform tabular models, which benefit from years of extensive engineering and domain-specific optimizations and inside knowledge. Therefore, several important questions arise: *(1) Can we integrate existing tabular models with generalist approaches (GNNs) to leverage their complementary strengths (speed, accuracy, and flexibility)? (2) Could such a hybrid framework*

Figure 1: Overview of the proposed hybrid modeling framework. The pipeline begins with feature-engineered tabular data processed by a tree-based model (e.g., LightGBM). Knowledge distillation generates additional representations, which are used as node features for the graph input to the static GNN in the training phase.

*simplify the assimilation of new data sources and development of simpler, accurate, and scalable models for new tasks?*

**Contributions.** Our work proposes a hybrid tabular-generalist modeling framework for predictive tasks on relational databases, which addresses the challenges of using existing tabular models while effectively dealing with temporal **scalability** in large relational databases. Specifically, the knowledge extracted from each existing tabular model is distilled into separate multi-layer perceptrons (MLPs). The embeddings generated by these MLPs are then integrated as additional features into a single static GNN. Having a single static GNN significantly reduces computational overhead compared to TGNNs used in prior work (Robinson et al., 2024). A diagram of the pipeline is shonw in figure 1. This modeling choice builds on the theoretical **time-then-graph** framework of Gao & Ribeiro (2022), which demonstrates that if a separate model captures the temporal dynamics of nodes and edges, the relational structure of a temporal graph can be fully captured by a static GNN. In our framework, the tabular models serve as the temporal model, followed by the application of a static GNN (details of this integration are provided in Section 3). This method not only improves efficiency but, as shown in experiments on the RelBench benchmark (Robinson et al., 2024), allows this lightweight model to outperform both standard tabular models and computationally expensive temporal GNNs models represented by RDL (Robinson et al., 2024).

*Our goal is not to suggest replacing generalist models like RDL with feature-engineered LIGHT-GBM distillations for TREELGNN. Instead, we aim to demonstrate how TREELGNN can build on tabular models already provided by industry-standard solutions, enhancing performance and enabling the seamless integration of unstructured data (e.g., images, text embeddings) into existing machine learning pipelines.*

## 2 BACKGROUND

In this section, we formally define the concept of relational databases and provide an overview of the predictive tasks and the primary methods used to address them. The notation adopted throughout this section roughly follows that of Fey et al. (2024) with changes that improve clarity in our settings.

### 2.1 RELATIONAL DATABASES

Relational databases provide a structured framework for organizing and managing interconnected data across multiple tables. Each table contains a collection of entities sharing a common schema, while relationships between tables capture complex interdependencies among these entities. A relational database can be formally defined as follows:

**Definition 1** (Relational Database). *A relational database $\mathcal{R}$ with $N$ tables is described through a set of entities $v \in \mathcal{V}$, where $v$ is a unique index of each entity in the dataset (a costumer, a product, a transaction, etc.). The $i$-th table in the database, $i \in [N]$, can be described as a set of rows representing each entity in the table: $\mathbb{T}_i = \{\{\boldsymbol{v}_v\}\}_{v \in \mathcal{V}:\phi(v)=i}$, where $\phi : \mathcal{V} \to [N]$ is a function that maps the entities to tables they belong to and $\boldsymbol{v}_v = (p_v, \boldsymbol{k}_v, \boldsymbol{x}_v, t_v)$ is row in the $i$-th table, where*

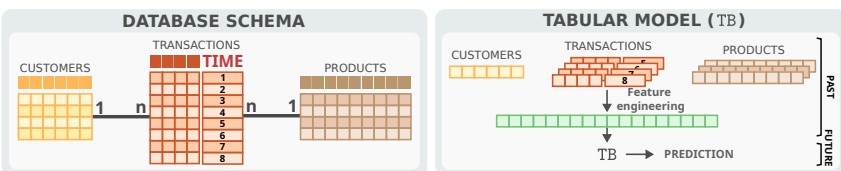

Figure 2: Relational Database schema and Features-engineered XGBoost.

- $p_v \in \mathbb{N}$ is the **primary key** that uniquely identifies the entity $v$;

- $\boldsymbol{k}_v = (k_v^1, \ldots, k_v^N)$ is the tuple of **foreign keys**, $k_v^c \in \mathbb{N}$, that allow to establish links between the tables, $c \in [N]$, with $k_v^c \equiv 0$ if $v$ does not refer to table $c$;

- $\boldsymbol{x}_v = (x_v^1, \ldots, x_v^{d_i})$ is the tuple of **features** of the entity $v$; entities often have features covering multiple modalities, where each feature $x_v^c$ can be categorical, numerical, or textual, $c \in [d_i]$;

- $t_v$ is a special feature if "$i$" is a fact-table, indicating the **timestamp** when entity $v$ appears.

*We also define the set of all "tables" as $\mathcal{T} = \{\mathbb{T}_1, \ldots, \mathbb{T}_N\}$. A link $(i,j)$ between tables $i, j \in [N]$ exists if a foreign key of $\mathbb{T}_i$ points to a primary key of $\mathbb{T}_j$, i.e., if exists $u \in \mathbb{T}_i$ and $v \in \mathbb{T}_j$ such that $k_u^j = p_v$. Let $\mathcal{L}$ be the set of such table links, more precisely, $\mathcal{L} = \{(i,j)|\forall i, j \in [N] \text{ s.t. } \exists u \in \mathbb{T}_i, \exists v \in \mathbb{T}_j, k_u^j = p_v\}$.*

Garcia-Molina (2008) classifies tables into two categories: *dimension tables*, which store descriptive attributes of entities, and *fact tables*, which capture interactions or events involving these entities. Dimension tables contain static features that remain stable over time, whereas fact tables record dynamic, time-dependent interactions, often using a special timestamp column to track temporal changes. The `rel-hm` database from RelBench (Robinson et al., 2024) is a representativa example of a relational database. It consists of customer purchase histories from the H&M e-commerce site and consists of three tables: (i) the **customer** table contains information customers, such as gender and birth year; (ii) the **product** provides details about products, including the description and the size; (iii) the **transaction** table records which customer buy a specific product. In this schema, the customer and product tables serve as dimension tables, as they store entities with immutable attributes, such as a customer's birth year. In contrast, the transaction table is classified as a fact table because it records interactions between customers and products. A representation of the `rel-hm` datproductsabase is provided in Figure 2, highlighting the table types and the relationships between tables.

## 2.2 ML TASKS IN RELATIONAL DATABASES

Many real-world machine learning tasks on relational databases consist on predicting the future state of specific entities. For instance, on `rel-hm`, one of the key tasks is forecasting the total sales of an article for the upcoming week. This is relevant to H&M as it enables effective inventory management, optimizes stock replenishment, and helps in crafting targeted marketing strategies to maximize revenue and reduce the risk of article shortages or overstock situations. Predictive tasks often involve entities from dimension tables and require the specification of a *seed time*, which is defined as "the present" in the prediction task. Consider the task of predicting the churn (no transactions) for a customer ($v$). Given a seed time $t$ in days ("the present"), we would like to predict the churn of $v$ over the "next week", i.e., in the interval $[t, t+7]$ in the database. Classical machine learning methods, such as gradient-boosted decision trees (GBDT), dominate relational database modeling due to their superior performance on these tabular data tasks (Gorishniy et al., 2021; Shwartz-Ziv & Armon, 2022). When paired with manually-designed feature engineering, tabular methods like LightGBM or XGBoost represent the industry's go-to methods for constructing predictive models on relational databases (Heaton, 2016). Unfortunately, improving these tabular models is both costly and time-consuming, particularly when incorporating new data sources or addressing new tasks. In recent years, the intrinsic structure of relational databases has attracted the use of generalist models such as TGNNs. We formalize next how a relational database can be

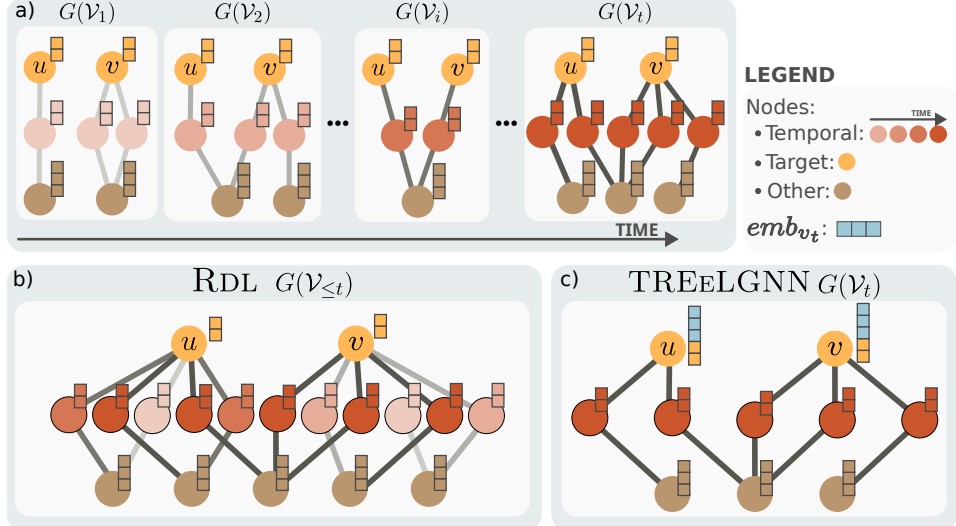

Figure 3: a) For each time $i$, the figure shows the graph $G(\mathcal{V}_i)$, which represents the interactions occurring at time $i$; b) graph $G(\mathcal{V}_{\leq t})$ is the aggregated graph consisting of all interactions from every time step up to $t$; this is the graph used by RDL, representing the cumulative interactions across all times; c) the graph $G(\mathcal{V}_t)$ in this panel represents only the interactions that occurred at time $t$, corresponding to the last graph in panel a), which is the training graph for TREELGNN.

represented as a temporal heterogeneous graph and recap how the recently proposed Relational Deep Learning (RDL) methods (Fey et al., 2024) exploit this representation for predictive tasks.

### 2.3 TGNNs FOR RELATIONAL DATABASE PREDICTION

We introduce the definition in Fey et al. (2024) of a relational database as a relational entity graph.

**Definition 2** (Relational Entity Graph). *For any subset of entities $\mathcal{V}' \subseteq \mathcal{V}$ in the* relational database $\mathcal{R}$ *we can construct a* heterogeneous graph $G(\mathcal{V}') = (\mathcal{V}', \mathcal{E}', \phi, \psi)$, *where $\mathcal{V}, \phi$ as given in Definition 1 and:*

1. $\mathcal{E}' = \{(v_1, v_2) \in \mathcal{V}' \times \mathcal{V}' \mid k_{v_2}^{\phi(v_1)} = p_{v_1}\}$ *is the set of edges between entities, which captures connections between nodes based on primary-foreign key relationships;*

2. $\psi : \mathcal{E}' \to \mathcal{L}$, *where $\mathcal{L} \subseteq [N] \times [N]$ is the* **edge type mapping** *function that assigns each edge $(v_1, v_2) \in \mathcal{E}'$ the pair of tables the belong to, i.e., $\psi(v_1, v_2) = (\phi(v_1), \phi(v_2))$;*

With the relational database represented as relational entity graph, predictions involving database entities can be redefined as node regression/classification tasks. In practice, tasks on relational databases are temporal and concern dimension nodes. For example, we may ask whether a customer will make another purchase at a given seed time $t + 1$.

**Time filtrations over $\mathcal{V}$: $\mathcal{V}_t$ and $\mathcal{V}_{\leq t}$.** It is therefore reasonable to restrict the relational entity graph to include only the nodes present at a certain time. We will denote the set of nodes and edges present at a given time respectively as $\mathcal{V}_t$ and $\mathcal{E}_t$, and the corresponding temporal relational entity graph as $G(\mathcal{V}_t)$. The relational entity graph can also be constructed for multiple time points; for example, we define $G(\mathcal{V}_{\leq t})$ as the relational entity graph that contains the nodes present at all time points up to $t$. Figure 3a shows several relational entity graphs at increasing time points, from 1 to $t$, Figure 3b shows $G(\mathcal{V}_{\leq t})$ resulting from the aggregation of all previous time points.

### 2.3.1 THE TGNN OF THE RDL MODEL (ROBINSON ET AL., 2024)

The RDL (Robinson et al., 2024) model employs a message-passing operator designed to account for both the heterogeneous and temporal nature of the graph $G$. Given a seed time $t \in \mathbb{R}$ and a $v \in \mathcal{V}$,

which serves as the task's target (for example predicting the churn for customer $v$ at time $t$) the relational entity graph $G(\mathcal{V}_{\leq t})$ is constructed (Figure 3b)). This graph serves as training examples for the heterogeneous GNN (Gilmer et al., 2017). While RDL presents a promising approach to leverage graph learning for predictive tasks on relational databases, it has a few notable drawbacks:

- In real-world applications, tabular models are refined over many years through extensive engineering and domain-specific optimizations. In contrast, generalist methods are unable to effectively use these extensive engineered features (Appendix E). This is a relevant limitation since it is known that GNNs tend to perform significantly better when initialized with features such as images or pre-trained embeddings, which provide a strong starting point for learning (Hu et al., 2020).

- The relational entity graph $G(\mathcal{V}_{\leq t})$ is used to make inferences for the next time interval. Thus, temporal information is handled by including all neighbors from previous time steps, ensuring that in the message-passing process, information flows only from earlier nodes to more recent ones. However, the use of RDL presents two challenges: first, there is no explicit mechanism (either recurrent or otherwise) to manage temporal dependencies; second, the graph $G(\mathcal{V}_{\leq t})$ tends to become large due to the aggregation of nodes and relations over all the times before $t$. We claim that it is possible to simultaneously: 1) better leverage temporal information by incorporating knowledge from tabular methods, and 2) construct a much lighter graph by utilizing the time-then-graph framework.

## 3 THE TREELGNN FRAMEWORK

We now propose the **TREELGNN** framework, a hybrid tabular-generalist modeling framework designed to integrate tabular data predictors (e.g., XGBoost, LIGHTGBM) with GNNs to capture both temporal and relational patterns. The key components of TREELGNN are a tabular model distillation and a time-then-graph representation, which we now describe in detail:

**(A) Tree Distillation:** Let $\mathbb{T}_i^{(\leq t)} = \{\{\boldsymbol{v}_v \in \mathbb{T}_i : \forall v \in \mathcal{V}_{\leq t}\}\}$ with $\mathbb{T}_i$ as in Definition 1. Let FE be a feature engineering process which assigns to each node $v \in \mathcal{V}_t$ a new $d_f$-dimensional feature based on all the entities of all the tables up to time $t$, i.e., $\boldsymbol{f}_{v,\leq t} = \text{FE}(v, \{\mathbb{T}_i^{(\leq t)}\}_{i=1}^N)$. A feature-engineered tabular model (e.g., XGBoost) is a model TB trained on the set $\mathcal{D} = \{(\boldsymbol{f}_{v,\leq t}, y_v) : v \in \mathcal{V}_{\leq t}\}$, where $y_v$ is the ground truth label associated to node $v \in \mathcal{V}_{\leq t}$ i.e., $y_v \in \mathcal{C} = \{c_1, \ldots, c_n\}$ for classification tasks or $y_v \in \mathbb{R}$ for regression. For all $v \in \mathcal{V}_{\leq t}$, we define $\hat{y}_v^{\text{TB}}(\boldsymbol{f}_{v,\leq t}) := \text{TB}(\boldsymbol{f}_{v,\leq t})$.

Our model distillation (Hinton, 2015) is performed via multi-task learning over an MLP with two task heads. The first task head is trained on the dataset $\mathcal{D} = \{(\boldsymbol{f}_{v,\leq t}, y_v) : v \in \mathcal{V}_{\leq t}\}$. The second task head is trained on the predictions produced by TB, i.e., $\hat{\mathcal{D}} = \{(\boldsymbol{f}_{v,\leq t}, \hat{y}_v^{\text{TB}}(\boldsymbol{f}_{v,\leq t})) : v \in \mathcal{V}_{\leq t}\}$.

Let $\hat{y}^{\text{MLP}}(\boldsymbol{f}_{v,\leq t}) := \text{MLP}^L(\boldsymbol{f}_{v,\leq t})$ be the softmax output of an $L$-layered MLP. The loss for the first task head is computed using the cross-entropy:

$$\mathcal{L}_{\text{hard}} = -\sum_{v \in \mathcal{V}_{\leq t}} \sum_{c \in \mathcal{C}} \mathbf{1}\{y_v = c\} \log(\hat{y}^{\text{MLP}}(\boldsymbol{f}_{v,\leq t}))_c, \tag{1}$$

where $(\hat{y}^{\text{MLP}}(\boldsymbol{f}_{v,\leq t}))_c$ is the probability that the MLP assigns to class $c$ for entity $v$. For computing the loss for the second task head, we need to soften the output from the tabular model using a *softmax with a temperature parameter* $F \geq 1$, which allows the tabular model to produce softer probability distributions over the classes $\mathcal{C}$, defined as follows:

$$p_c^{\text{TB},F}(\boldsymbol{f}_{v,\leq t}) = \text{softmax}((\log \hat{y}_v^{\text{TB}}(\boldsymbol{f}_{v,\leq t}))/F), \tag{2}$$

where $F$ is the temperature parameter that controls the smoothness of the output distribution. The distillation loss is then computed as the cross-entropy between the soft labels provided by the tabular model and the soft labels generated by the MLP. This can be expressed as

$$\mathcal{L}_{\text{soft}} = -\sum_{v \in T} \sum_{c \in \mathcal{C}} p_c^{\text{TB},F}(\boldsymbol{f}_{v,\leq t}) \log(\hat{y}^{\text{MLP}}(\boldsymbol{f}_{v,\leq t}))_c. \tag{3}$$

The total loss for the distillation process is a weighted combination of these two losses:

$$\mathcal{L}_{\text{total}} = \alpha \mathcal{L}_{\text{hard}} + (1 - \alpha) F^2 \mathcal{L}_{\text{soft}}, \tag{4}$$

where $\alpha$ is a hyperparameter that controls the trade-off between the ground truth learning and the distillation learning, and $F$ is the temperature parameter used to soften the predictions from the tabular model. In the case of regression tasks, we follow a similar procedure but replace the cross-entropy losses with an appropriate regression loss, i.e., the mean absolute error (MAE). Once the MLP is trained, the embedding generated by the last hidden layer, $emb_{v,\leq t} = \text{MLP}^{L-1}(\boldsymbol{f}_{v,\leq t})$ contains the knowledge learned from the highly optimized features engineered tabular methods. These embeddings can be integrated as additional node features. In principle, any existing tabular model from the literature can be used as the TB component; for our experiments we follow (Robinson et al., 2024) and choose LIGHTGBM.

**(B) Time-then-graph representation:** The core idea behind TREELGNN is to use a static GNN on a graph built only with the nodes and relations at time $t$ to infer the next time step. The node features are integrated with embeddings obtained through distillation, which effectively capture temporal dynamics.

Given a seed time $t \in \mathbb{R}$ and node $v \in \mathcal{V}$ which serves as the task's target, TREELGNN constructs the relational entity graph using only the nodes from that specific time, i.e., $G(\mathcal{V}_t)$ (Figure 3c). In this way, the graph is significantly smaller because only the interactions between entities occurring at that specific time are considered. To incorporate historical temporal information, the embeddings obtained through distillation are integrated as additional feature in the graph nodes as follows:

$$\boldsymbol{h}_v^0 = f(\boldsymbol{x}_v, emb_{v,\leq t}), \forall v \in \mathcal{V}_t \text{ s.t. } \phi(v) = i^*, \tag{5}$$

where $f$ is a function that aggregates the original features and the additional ones (in our experiments we used $f$ as the concatenation), $\boldsymbol{x}_v$ are the node features as defined in Definition 1 and $i^*$ is such that $\mathbb{T}_{i^*}$ is the table whose entities are the target of the task (e.g customer in the user-churn task).

The core of TREELGNN is a static heterogeneous GNN; for every heterogeneous message-passing layer $\ell \leq L'$:

$$\boldsymbol{h}_v^{\ell+1} = \text{COMB}_{\phi(v)}\left(\boldsymbol{h}_v^\ell, \left\{\text{AGG}_{(\phi(v),j)}\left(\{\boldsymbol{h}_u^\ell : u \in \mathcal{N}_j^{(t)}(v)\}\right) : \forall j \in [N], |\mathcal{N}_j^{(t)}(v)| > 0\right\}\right), \tag{6}$$

where

$$\mathcal{N}_j^{(t)}(v) = \{u : (v,u) \in \mathcal{E}_t, \phi(u) = j\}, \tag{7}$$

denotes the set of neighbors of $v$ at time $t$ that belong to the $j$-th table, $\phi$ is defined as in Definition 1, and $\mathcal{E}_t = \mathcal{E}'$ is as given in Definition 2 for $\mathcal{V}' = \mathcal{V}_t$. Aggregation is performed separately for each target-neighbor table pair $(\phi(v),j)$ using a neural aggregation function $\text{AGG}_{(\phi(v),j)}$ which is parameterized by the specific table pair. Subsequently, a neural combination function $\text{COMB}_{\phi(v)}$, parameterized by the type of the target node, merges the target node's previous layer representation with the aggregated representations of its neighbors across all relations.

In order to perform the COMB and AGG operations in Equation (6), any functions borrowed from heterogeneous static GNNs can be used. We opted for HeteroGraphSAGE (Hamilton et al., 2017) since it is the same message-passing layer used in RDL, allowing for a fairer comparison in our experiments. It is worth mentioning that the use of a static GNN combined with node embeddings to handle temporality ($emb_{v,\leq t}$ in our case) is motivated by the theoretical results on the *time-then-graph* framework in Gao & Ribeiro (2022). These results prove that when temporal information is effectively encoded as node/edge embeddings, a static GNN can achieve as strong performance on temporal graph tasks as the best temporal GNNs, if not better under some scenarios. Our approach indeed falls in the family of time-then-graph methods for temporal graphs.

**Expressiveness of TREELGNN.** In light of Chen & Wang (2024), we can draw conclusions about the expressiveness of our method and compare it to RDL, specifically in terms of logical expressiveness in binary classification, which is a prevalent task in many relatable datasets. In TREEL-GNN, the heterogeneous GNN used as the model core is a HeteroGraphSAGE. This model has been proven capable of capturing every Boolean node classifier expressible in $\mathcal{FOC}_2$ logic. $\mathcal{FOC}_2$, which stands for First-Order Logic with Counting Quantifiers, is a formal system that extends first-order logic by allowing quantification over sets and counting. Using Theorem 17 in Chen & Wang (2024), since the graphs constructed from relational databases are bounded and simple, we can state that: 1) RDL is incomparable to the time-and-graph framework (Gao & Ribeiro, 2022) in terms of the

Table 1: Entity regression results (MAE, lower is better). Best values are in bold. See Table 8 in Appendix D for standard deviations. TREELGNN demonstrates significantly superior performance with respect to the baselines, with gains over RDL ranging from 0.6% to 33.9%, and gains over LIGHTGBM ranging from 2.6% to 4.7%.

| | | LIGHTGBM | RDL | RDL w. P. | RDL w. D. | TREELGNN | Gain wrt RDL (%) | Gain wrt LIGHTGBM (%) |
|---|---|---|---|---|---|---|---|---|
| rel-f1 | **driver-position** | 4.010 | 4.142 | 3.991 | 4.120 | **3.861** | +6.8 | +3.7 |
| rel-hm | **item-sales** | 0.038 | 0.056 | 0.050 | 0.052 | **0.037** | +33.9 | +2.6 |
| rel-event | **user-attendance** | 0.249 | 0.255 | 0.248 | 0.247 | **0.238** | +6.7 | +4.4 |
| rel-stack | **post-votes** | 0.068 | 0.065 | 0.065 | 0.065 | **0.064** | +0.6 | +4.7 |
| rel-amazon | **user-ltv** | 14.212 | 14.314 | 14.187 | 13.974 | **13.587** | +5.1 | +4.4 |
| | **item-ltv** | 49.917 | 50.053 | 49.189 | 48.752 | **48.112** | +3.8 | +3.6 |
| *GNN* | | ✗ | ✓ | ✓ | ✓ | ✓ | | |
| *Time-**then**-graph* | | - | ✗ | ✗ | ✗ | ✓ | | |

logical expressiveness; 2) by incorporating temporal handling through additional features distilled from LIGHTGBM, TREELGNN becomes strictly more expressive than RDL.

## 4 RELATED WORK

Relational databases are integral to a wide-range of applications, from e-commerce platforms (Agrawal et al., 2001) and social media networks (Almabdy, 2018) to banking systems (Aditya et al., 2002) and healthcare services (Park et al., 2014). Tree-based methods, especially XGBoost, remain the preferred methods for learning on these relational data (Fey et al., 2024). Indeed, although efforts to design deep learning architectures for tabular data has shown promising results (Huang et al., 2020; Arik & Pfister, 2021; Gorishniy et al., 2021; 2022; Chen et al., 2023), no deep-learning model has yet been demonstrated to clearly outperform tree-based methods on tabular data (Shwartz-Ziv & Armon, 2022; McElfresh et al., 2024).

Among the deep learning models proposed for relational data, graph neural networks (GNNs) have also gained attention, with models specifically designed to handle relations among nodes (Schlichtkrull et al., 2018; Cvitkovic, 2020; Zahradník et al., 2023; Ferrini et al., 2024). Recently, (Fey et al., 2024; Robinson et al., 2024) proposed a general end-to-end learnable framework for solving tasks on relational data that incorporates a temporal dimension. This proposed approach bridges the gap between relational GNNs and temporal GNNs (Kapoor et al., 2020; Gao et al., 2021; Sankar et al., 2020; Cui et al., 2019; Zhao et al., 2019; Lv et al., 2020; Pareja et al., 2020; Jin et al., 2020; Manessi et al., 2020; Rossi et al., 2020; Gao & Ribeiro, 2022; Heeg & Scholtes, 2023; Longa et al., 2023; von Pichowski et al., 2024; Marisca et al., 2024; Beddar-Wiesing et al., 2024), introducing RDL which serves as both our starting point and main competitor in the development of more effective and light-way GNN-based solutions for relational databases.

Other works have explored the combination of GNNs with boosting methods (Ivanov & Prokhorenkova, 2021; Sun et al.; Shi et al., 2021; Zheng et al., 2021; Tang et al., 2024; Deng et al., 2021).However, they focus on improving standard GNNs for graph datasets that are not derived from relational databases, and therefore lack temporal and heterogeneous components. Moreover, while these methods often aim to replace trees with GNNs within a boosting setup, we instead incorporate pretrained tree-based models as a dedicated component for modeling temporality. This is complemented by a static GNN, which captures the structural relationships within the data.

## 5 RESULTS

We now show the effectiveness of TREELGNN against state-of-the-art baselines across multiple experimental settings. The experiments primarily seek to show that TREELGNN significantly reduces both training and inference times without sacrificing accuracy. In all scenarios, TREELGNN not only maintains competitive accuracy but outperforms the best baselines in many scenarios, all

while being considerably faster. A detailed description of the RelBench datasets used is provided in Appendix A. Detailed model configurations and training procedures are provided in Appendix B.

*Before continuing to our experiments, it is important to reemphasize that our goal is not to suggest replacing generalist models like RDL with feature-engineered LIGHTGBM distillations for TREELGNN. Instead, these experiments seek to show that organizations with existing high-performing tabular models (e.g., XGBoost) can leverage TREELGNN to enhance their performance and seamlessly integrate new features and unstructured data (e.g., images, text embeddings) into their machine learning pipelines.*

## 5.1 DATASETS

For these experiments we consider four datasets from the RelBench benchmark (Robinson et al., 2024). The `rel-hm` relational database contains customer purchase histories and product metadata from the brand's online shopping network. The `rel-f1` database provides comprehensive data on Formula 1 racing since 1950, including information about drivers, constructors, circuits, and race results. The `rel-event` database is derived from the Hangtime app and includes anonymized user actions, event metadata, and social relations data. The `rel-stack` dataset is sourced from the stats-exchange site, comprising user activity, posts, comments, and voting histories. Finally, `rel-amazon` stores information about product, users and reviews from Amazon platform. For each dataset, we considered several tasks, which include both regression and classification tasks aimed at making future predictions at the entity level of the tables.

## 5.2 TREELGNN CONFIGURATION AND BASELINES

TREELGNN is configured using two models: (a) the pretrained feature-engineered LIGHTGBM model from RelBench (Robinson et al., 2024); and (b) a static GNN (HGSAGE) that is a static HeteroGraphSAGE (Hamilton et al., 2017), identical to the (static) GNN used at each timestep in the RDL model (Robinson et al., 2024).

We evaluate TREELGNN against two baselines which are the strongest-performing methods in Robinson et al. (2024): the pretrained feature-engineered LIGHTGBM model from RelBench and the RDL model. Since TREELGNN uses the additional features derived through distillation, to ensure a fair comparison, we also create two versions of RDL that incorporate these additional features. First, we added the pointwise predictions of LIGHTGBM as extra features on the nodes (RDL w.P.). Second, we introduced the embeddings obtained from the distillation process, as done with TREELGNN (RDL w.D.). The architectural details of the different models are provided in Appendix B, the specifics of the distillation process can be found in Appendix C, while the details about the batch size used for calculating runtime can be found in Appendix F.4.

## 5.3 EXPERIMENTAL RESULTS

**Regression tasks (MAE).** Figure 4(a) shows TREELGNN achieves between +0.6% to +34% lower test errors than RDL, but it is also significantly faster in training and inference times compared to RDL across all regression tasks, demonstrating speedups between $100\times$ to $1300\times$ in train and between $35\times$ to $1050\times$ in inference, which we will discuss later.

Table 1 shows the performance of TREELGNN against other baselines, more specifically, against LIGHTGBM and all versions of RDL: RDL original, RDL with the predicted $y$ from LIGHTGBM (RDL w. P.), and RDL with the LIGHTGBM-distilled embeddings $emb_{v,\leq t}$ (RDL w. D.). We see that RDL does not gain very much from incorporating the LIGHTGBM predictions, possibly due to the size of its temporal graph ($G(\mathcal{V}_{\leq t})$ in Figure 2) interfering with extracting information from the extra LIGHTGBM-related features. We also note that the gains of TREELGNN over LIGHTGBM are more modest, they range from +2.6% to +4.7%.

**Classification tasks (AUCROC).** Figure 5(a) shows that TREELGNN outperforms RDL in four dataset with gains ranging +1.2% to +7.9% and TREELGNN underperforms RDL in three datasets with modest losses ranging from -2.1% to -0.8%, but noting that in the tasks TREELGNN loses to RDL, it is between $100\times$ to $2800\times$ faster in training and $95\times$ to $500\times$ faster in inference than RDL.

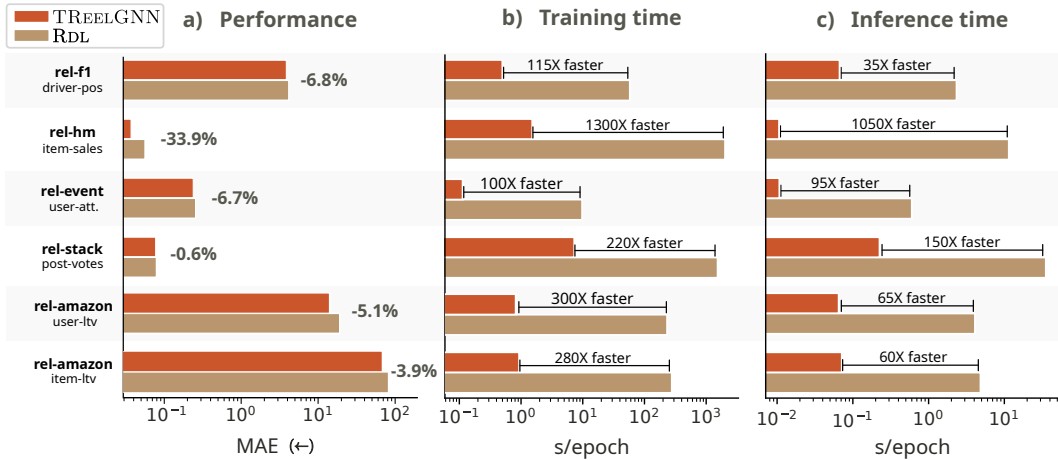

Figure 4: Mean Average Error (MAE) –lower is better— (a), training time (b) and inference time (c) for regression tasks. TREELGNN achieves between 0.6% to 34% lower test errors than RDL, but it is also significantly faster in training and inference times compared to RDL across all regression tasks, demonstrating speedups between 100× to 1300× in train and between 35× to 1050× in inference.

Table 2: Entity classification results (AUROC, higher is better). Best values are in bold. See Table 10 in Appendix D for standard deviations. TREELGNN achieve good performance in classification; even if it does not always outperform competitors, the average performance gain wrt LIGHTGBM and RDL are 1,4% and 1,6% respectively.

| | | LIGHTGBM | RDL | RDL w. P. | RDL w. D. | TREELGNN | Gain wrt RDL (%) | Gain wrt LIGHTGBM (%) |
|---|---|---|---|---|---|---|---|---|
| rel-f1 | **driver-dnf** | 70.52 | 71.08 | 69.93 | 71.57 | **73.55** | **+3.5** | **+4.3** |
| | **driver-top3** | 82.77 | 80.30 | 82.28 | 83.28 | **84.73** | **+5.5** | **+2.4** |
| rel-hm | **user-churn** | 69.12 | 69.09 | 69.24 | **69.56** | 68.93 | -0.9 | -0.3 |
| rel-event | **user-ignore** | 82.62 | 77.82 | 70.72 | 79.13 | **83.98** | **+7.9** | **+1.6** |
| | **user-repeat** | 75.78 | 76.50 | 76.57 | 76.63 | **77.77** | **+1.7** | **+2.6** |
| rel-stack | **user-engagement** | 90.34 | 90.59 | **90.66** | 90.50 | 89.02 | -1.7 | -1.5 |
| | **user-badge** | 86.23 | **88.54** | 88.42 | 88.57 | 86.71 | -2.1 | **+0.1** |
| rel-amazon | **user-churn** | 68.34 | **70.42** | 69.81 | 69.90 | 69.87 | -0.8 | **+2.2** |
| | **item-churn** | 82.62 | 82.81 | 82.93 | 83.12 | **83.84** | **+1.2** | **+1.5** |
| | *GNN* | ✗ | ✓ | ✓ | ✓ | ✓ | | |
| | *Time-**then**-graph* | - | ✗ | ✗ | ✗ | ✓ | | |

Table 2 shows the performance of TREELGNN against other baselines, more specifically, against LIGHTGBM and all versions of RDL: RDL original, RDL w. P., and RDL w.D. The performance gain of TREELGNN with respect to LIGHTGBM is between -1.5% to +4.3%. These findings confirm that TREELGNN is a robust and competitive model, excelling in most cases and losing only marginally in others. Notably, RDL w.P. and RDL w.D. achieve better performance compared to RDL, but they still underperform with respect to TREELGNN. We note that, in three tasks, RDL w.P show lower performance than LIGHTGBM. This may be because the node features in the user-ignore task are quite large, and adding just a single value for the prediction is insufficient hint for the model to understand its importance.

**Comparing training and inference times.** TREELGNN *achieves a substantial reduction in training times* compared to RDL across all datasets and tasks, with speedups ranging from 100x to 2800x (see Figure 4 (b) and Figure 5 (b)). This dramatic speedup is due primarily to two factors: (i) the effect of the number of timestamps of the tasks, which does not affect TREELGNN and significantly affects RDL (see Figure 2) and (ii) the much smaller number of model parameters in TREELGNN (see Appendix B). Unlike RDL, where the training graph $G(\mathcal{V}_{\leq t})$ is constructed using

entities and relations of all the time up to $t$, TREELGNN's graph $G(\mathcal{V}_t)$ only uses the information at the timestamp $t$ before the inference, so that the size of its graph is independent of the number of snapshots.

More importantly for organization that already have high-performing tabular models (e.g., XG-Boost), **incorporating TREELGNN is substantially faster than using temporal GNNs such as RDL at inference time**, achieving speedups ranging from $15\times$ to $1050\times$ across various tasks (See Figure 4 (c) and Figure 5 (c)). These speedups largely compensate for the occasional limited drop in classification performance (Table 2). Indeed, the largest decrease in classification performance is 2.1% in the user-badge task of the rel-stack dataset, where TREELGNN has a 95-fold increase in inference speed over RDL. This substantial improvement makes TREELGNN particularly well-suited for relational database ML applications, that demand low-latency and low-computational overhead. As a result, TREELGNN is not only preferred for scenarios requiring rapid response times but is also highly practical for deployment in industrial settings where pretrained tabular models already exist and computational resources and inference latency are critical constraints.

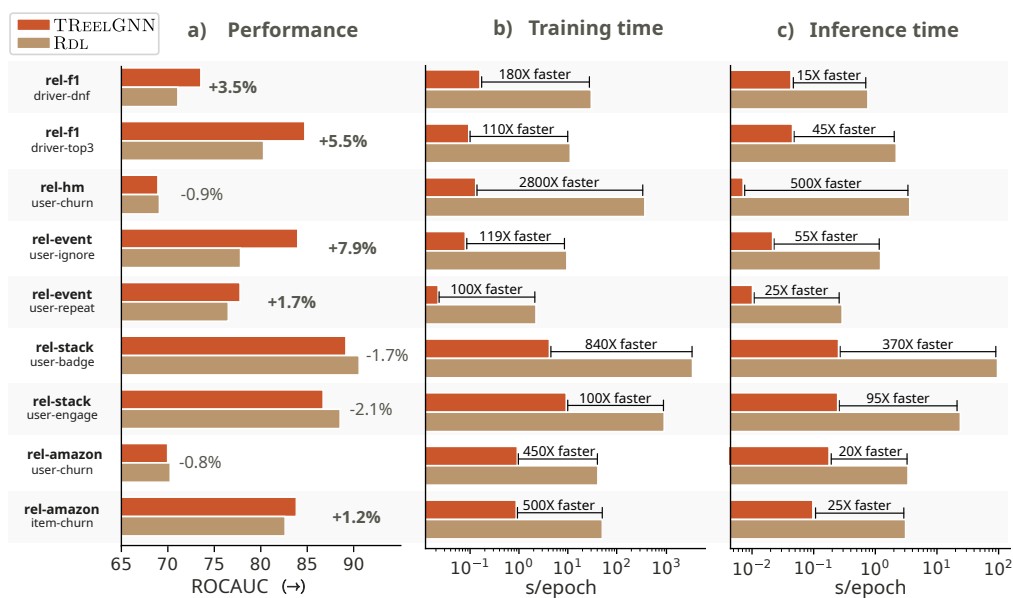

Figure 5: AUCROC (a), training time (b) and inference time (c) for classification tasks. TREEL-GNN is significantly faster than RDL, achieving speedups ranging from $100\times$ to $2800\times$ in training and from $15\times$ to $500\times$ in inference. While TREELGNN mildly underperforms RDL in three tasks, with a maximum performance loss of 2%, this is compensated by a training speed that is at least $100\times$ faster and inference speed that is at least $25\times$ faster in these same scenarios.

## 6 CONCLUSION

In this work, we introduced the TREELGNN framework, a novel relational deep learning method that integrates tabular models and graph neural networks in a time-then-graph framework. Our results on the RelBench benchmark demonstrate that leveraging the strengths of both tabular models and static GNNs can significantly improve predictive accuracy and efficiency compared to using either approach in isolation. Specifically, by embedding predictive features from existing tabular models into a unified static GNN framework, we achieve substantial performance gains across multiple tasks at a fraction of the computational cost of current RDL approaches.

The TREELGNN framework simplifies the integration of new data into predictive workflows using a static GNN, enabling real-time inference and enhancing its practicality for industrial applications. Additionally, this work demonstrates the potential of graph representation learning to complement traditional tabular models, paving the way for future research on hybrid architectures that effectively combine feature-engineered and graph-based representations for complex relational data.

REPRODUCIBILITY STATEMENT

Additional information regarding the experiments and implementation details, as well as source code are provided in Appendix B.

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

# A DATASET

The Relational Deep Learning Benchmark (RelBench) is a collection of large-scale, real benchmark datasets for machine learning on relational databases. We consider four databases of RelBench and the respective predictive tasks:

**rel-event** originates from the Hangtime mobile app, which tracks users' social plans and interactions with friends. The dataset contains anonymized data on user actions, event metadata, and demographic information, as well as users' social connections, allowing for an analysis of how these relations influence behavior. No personally identifiable information is included in the dataset. The entity predictive tasks on this database are:

- **user-attendance**: Predict how many events each user will respond "yes" or "maybe" to within the next seven days.
- **user-repeat**: Determine whether a user will attend another event (by responding "yes" or "maybe") within the next seven days, given they have attended an event in the last 14 days.
- **user-ignore**: Predict whether a user will ignore more than two event invitations within the next seven days.

**rel-f1** comprises historical data and statistics from Formula 1 racing, covering the period from 1950 to the present. It includes comprehensive information on key stakeholders, such as drivers, constructors, engine manufacturers, and tire manufacturers. The dataset highlights geographical details of circuits, along with detailed historical season data, including race results, practice sessions, qualifying positions, sprints, and pit stops. The entity predictive tasks on this database are:

- **driver-position**: Forecast the average finishing position of each driver across all races in the upcoming two months.
- **driver-dnf**: Predict whether a driver will fail to complete a race (DNF - Did Not Finish) within the next month.
- **driver-top3**: Determine whether a driver will qualify within the top 3 positions in a race over the next month.

**rel-hm** comprises extensive customer and product data from the company's online shopping platform of H&M. It includes detailed customer purchase histories and a wide range of metadata, covering everything from customer demographics to comprehensive product information. This dataset enables a deep analysis of shopping behavior across a broad network of brands and stores. The entity predictive tasks on this database are:

- **user-churn**: Predict customer churn (i.e., no transactions) within the next week.
- **item-sales**: Estimate the total sales for a product (summed over the associated transactions) during the next week.

**rel-stack** captures detailed interactions from the network of question-and-answer websites Stack Exchange. It includes comprehensive records of user activity, such as biographies, posts, comments, edit histories, voting patterns, and links between related posts. The reputation system within Stack Exchange enables self-moderation of the community. In our experiments, we use data from the stats-exchange site. The entity predictive tasks on this database are:

- **user-engagement**: Predict whether a user will engage (e.g., through votes, posts, or comments) within the next three months.
- **post-votes**: Forecast how many votes a user's post will receive over the next three months.
- **user-badge**: Predict if a user will be awarded a new badge during the next three months.

**rel-amazon** The Amazon E-commerce database documents products, users, and reviews from Amazon's platform, providing comprehensive details about products and their associated reviews. Each product entry includes its price and category, while reviews capture the overall rating, whether

Table 3: Tasks details.

| Dataset | Task name | Task type | #Rows Train | #Rows Val | #Rows Test | #Unique Entities | %train/test Entity Overlap |
|---|---|---|---|---|---|---|---|
| rel-event | **user-attendance** | entity-reg | 19 261 | 2 014 | 2 006 | 9 694 | 14.6 |
| | **user-repeat** | entity-cls | 3 842 | 268 | 246 | 1 514 | 11.5 |
| | **user-ignore** | entity-cls | 19 239 | 4 185 | 4 010 | 9 799 | 21.1 |
| rel-f1 | **driver-dnf** | entity-cls | 11 411 | 566 | 702 | 821 | 50.0 |
| | **driver-top3** | entity-cls | 1 353 | 588 | 726 | 134 | 50.0 |
| | **driver-position** | entity-reg | 7 453 | 499 | 760 | 826 | 44.6 |
| rel-hm | **user-churn** | entity-cls | 3 871 410 | 76 556 | 74 575 | 1 002 984 | 89.7 |
| | **item-sales** | entity-reg | 5 488 184 | 105 542 | 105 542 | 1 005 542 | 100.0 |
| rel-stack | **user-engagement** | entity-cls | 1 360 850 | 85 838 | 88 137 | 88 137 | 97.4 |
| | **user-badge** | entity-cls | 3 386 276 | 247 398 | 255 360 | 255 360 | 96.9 |
| | **post-votes** | entity-reg | 2 453 921 | 156 216 | 160 903 | 160 903 | 97.1 |
| rel-amazon | **user-churn** | entity-cls | 4 732 555 | 409 792 | 351 885 | 1 585 983 | 88.0 |
| | **item-churn** | entity-cls | 2 559 264 | 177 689 | 166 842 | 416 352 | 93.1 |
| | **user-ltv** | entity-reg | 4 732 555 | 409 792 | 351 885 | 1 585 983 | 88.0 |
| | **item-ltv** | entity-reg | 2 707 679 | 166 978 | 178 334 | 427 537 | 93.5 |

the reviewer purchased the product, and the review text. For our analysis, we focus specifically on a subset of book-related products.

- **user-churn**: Predict whether a user will refrain from reviewing any products in the next three months (1 for no reviews, 0 otherwise).

- **item-churn**: Determine if a product will receive no reviews in the next three months (1 for no reviews, 0 otherwise).

- **user-ltv**: Predict the value of the total products a user purchases and reviews over the next three months.

- **item-ltv**: Predict the value of the total purchases and reviews a product receives in the next three months.

Further details regarding the tasks are provided in Table 3.

## B EXPERIMENTAL DETAILS

The hyperparameter search was performed using grid search, exploring values for the learning rate $(0.1, 0.01, 0.001)$, dropout rates $(0.1, 0.2, 0.3, 0.4, 0.5)$, and the number of layers (ranging from 2 to 6). Our method is implemented with PyTorch, PyTorch Geometric (Fey & Lenssen, 2019), and TorchFrame (Hu et al., 2024), and the experiments were conducted on a single RTX-4090 GPU with 24GB of memory. For classification tasks, we used the BCEWithLogitsLoss function, and for regression tasks, we employed L1Loss. The source code for reproducibility is available at https://anonymous.4open.science/r/TReeLGNN-3DED/README.md.

Table 4 presents the number of parameters of TREELGNN and RDL across the different tasks.

## C DISTILLATION

The knowledge distillation from LIGHTGBM into an MLP was carried out as described in Section Section 3. A grid search was performed to tune the learning rate, dropout, the $\alpha$ value, temperature $T$, and the number of layers. The size of the penultimate layer, from which the embeddings are extracted, was fixed at 10. The results of the distillation for the classification tasks are presented in Table 6, where the first column shows the AUC-ROC against the true target, and the second column shows the AUC-ROC against the predictions of LIGHTGBM. In Table 5, the results for the regression tasks are also reported.

Table 4: Number of parameters. TREELGNN has significantly fewer parameters than RDL.

|  | Dataset | Task | RDL | TREELGNN |
|---|---|---|---|---|
| Classification | `rel-f1` | **driver-dnf** | 5 073 793 | 271 803 |
| | | **driver-top3** | 5 073 793 | 648 395 |
| | `rel-hm` | **user-churn** | 2 178 945 | 22 204 |
| | `rel-event` | **user-ignore** | 5 942 785 | 1 231 871 |
| | | **user-repeat** | 5 942 785 | 467 583 |
| | `rel-stack` | **user-engagement** | 4 322 177 | 2 847 127 |
| | | **user-badge** | 4 322 177 | 3 586 454 |
| | `rel-amazon` | **user-churn** | 5 129 348 | 1 622 173 |
| | | **item-churn** | 5 129 348 | 1 730 512 |
| Regression | `rel-f1` | **driver-position** | 5 073 793 | 1 372 395 |
| | `rel-hm` | **item-sales** | 2 178 945 | 73 409 |
| | `rel-event` | **user-attendance** | 5 942 785 | 1 518 271 |
| | `rel-stack` | **post-votes** | 4 322 177 | 1 987 955 |
| | `rel-amazon` | **user-ltv** | 5 129 348 | 925 571 |
| | | **item-ltv** | 5 129 348 | 1 113 729 |

Table 5: Distillation results for the regression tasks in MAE with respect to the real target and to the LIGHTGBM prediction.

| Dataset | Task | MAE vs Real | MAE vs LIGHTGBM |
|---|---|---|---|
| `rel-f1` | **driver-position** | 3.881 | 2.411 |
| `rel-hm` | **item-sales** | 0.040 | 0.022 |
| `rel-event` | **user-attendance** | 0.269 | 0.068 |
| `rel-stack` | **post-votes** | 0.067 | 0.007 |
| `rel-amazon` | **user-ltv** | 14.438 | 7.319 |
| | **item-ltv** | 50.925 | 32.264 |

## D  DETAILED PERFORMANCE

The experiments reported in Section 5 were conducted over 5 runs with 5 different seeds. The complete results are presented in the following tables. Table 7 shows the mean and standard deviation of the validation performance for the baselines and TREELGNN on the regression tasks; Table 8 presents the mean and standard deviation of the test performance for the baselines and TREELGNN on the regression tasks. Table 9 and Table 10 present the corresponding results for the classification tasks.

## E  TREELGNN W/O TIME WITH FEATURE ENGINEERING

We wanted to test the hypothesis that even when generalist models are allowed to use these engineered features, the performance remains suboptimal, underscoring the fact that such features are specifically tailored for tabular models. We conducted a preliminary experiment on the **driver-top3** task of the `re-f1` dataset, where the same engineered features were directly applied as node features in the graph without employing any tabular methods (TREELGNN W/O TIME with F.E.). The poor performance confirm that the feature produced by the feature engineering are primarily designed for tabular models.

Table 6: Distillation results for the classification tasks in AUCROC with respect to the real target and to the LIGHTGBM prediction.

| Dataset | Task | AUCROC vs Real | AUCROC vs LIGHTGBM |
|---|---|---|---|
| rel-f1 | driver-top3 | 82.74 | 89.92 |
| | driver-dnf | 79.32 | 90.17 |
| rel-hm | user-churn | 69.81 | 82.03 |
| rel-event | user-ignore | 80.23 | 91.08 |
| | user-badge | 85.77 | 92.01 |
| rel-stack | user-engage | 87.05 | 89.41 |
| rel-amazon | user-churn | 66.93 | 89.32 |
| | item-churn | 79.91 | 89.73 |

Table 7: Validation MAE with standard deviation over 5 runs.

| | | LIGHTGBM | RDL | RDL w. P. | RDL w. D. | TREELGNN |
|---|---|---|---|---|---|---|
| rel-f1 | driver-pos | $2.800 \pm 0.030$ | $3.130 \pm 0.050$ | $2.830 \pm 0.050$ | $3.130 \pm 0.030$ | $2.910 \pm 0.070$ |
| rel-hm | item-sales | $0.048 \pm 0.001$ | $0.065 \pm 0.001$ | $0.060 \pm 0.000$ | $0.061 \pm 0.001$ | $0.046 \pm 0.000$ |
| rel-event | user-attend. | $0.249 \pm 0.002$ | $0.246 \pm 0.004$ | $0.243 \pm 0.004$ | $0.244 \pm 0.003$ | $0.244 \pm 0.003$ |
| rel-stack | post-votes | $0.062 \pm 0.001$ | $0.059 \pm 0.001$ | $0.059 \pm 0.003$ | $0.059 \pm 0.008$ | $0.059 \pm 0.002$ |
| rel-amazon | user-ltv | $11.482 \pm 0.001$ | $12.132 \pm 0.007$ | $12.112 \pm 0.001$ | $11.892 \pm 0.002$ | $11.325 \pm 0.025$ |
| | item-ltv | $44.314 \pm 0.001$ | $45.140 \pm 0.068$ | $44.910 \pm 0.013$ | $44.201 \pm 0.025$ | $43.121 \pm 0.078$ |

# F ADDITIONAL EXECUTION TIME COMPARISON

## F.1 PREPROCESSING

Table 12 compares the graph construction times of TREELGNN model and RDL. For TREELGNN model, preprocessing involves three steps: first, computing predictions with LIGHTGBM; second, distilling these predictions into embeddings; and finally, constructing static graphs. In contrast, RDL requires the construction of a temporal graph and the embedding of input features. While the preprocessing time for TREELGNN model can be up to twice that of RDL, it is performed only once. This preprocessing leads to a significantly faster training process compared to RDL, making the overall pipeline more efficient.

## F.2 END-TO-END TRAINING TIME

This section presents the end-to-end training results for TREELGNN model, RDL, and LIGHTGBM. All models were trained using early stopping, and the reported results are averaged over five runs. The table shows that TREELGNN model is significantly faster than RDL and generally outperforms it by 4.74%. In contrast, the training time of TREELGNN model is comparable to that of LIGHTGBM, but it achieves an average performance improvement of 2.4%.

## F.3 INFERENCE TIME

## F.4 BATCH SIZE

In this section, we explain the mini-batching strategy used by TREELGNN model compared to that of RDL. In the case of TreeLGNN, a batch of size 1 corresponds to the entire graph at the current timestamp, as the model processes the full set of interactions at that time. In contrast, for RDL, a batch of size 1 refers to a single node along with its temporal neighborhood. Given these differences

Table 8: Test MAE with standard deviation over 5 runs.

| | | LIGHTGBM | RDL | RDL w. P. | RDL w. D. | TREELGNN |
|---|---|---|---|---|---|---|
| rel-f1 | **driver-pos** | $4.010 \pm 0.080$ | $4.142 \pm 0.110$ | $3.991 \pm 0.120$ | $4.120 \pm 0.270$ | $3.861 \pm 0.045$ |
| rel-hm | **item-sales** | $0.038 \pm 0.001$ | $0.056 \pm 0.001$ | $0.050 \pm 0.000$ | $0.052 \pm 0.000$ | $0.037 \pm 0.000$ |
| rel-event | **user-attend.** | $0.249 \pm 0.003$ | $0.255 \pm 0.004$ | $0.248 \pm 0.002$ | $0.247 \pm 0.001$ | $0.238 \pm 0.003$ |
| rel-stack | **post-votes** | $0.068 \pm 0.000$ | $0.065 \pm 0.000$ | $0.065 \pm 0.000$ | $0.065 \pm 0.000$ | $0.064 \pm 0.000$ |
| rel-amazon | **user-ltv** | $14.210 \pm 0.000$ | $14.313 \pm 0.013$ | $14.183 \pm 0.0382$ | $13.9712 \pm 0.010$ | $13.582 \pm 0.043$ |
| | **item-ltv** | $49.912 \pm 0.000$ | $50.052 \pm 0.163$ | $49.181 \pm 0.063$ | $48.751 \pm 0.023$ | $48.115 \pm 0.059$ |

Table 9: Validation AUCROC with standard deviation over 5 runs.

| | | LIGHTGBM | RDL | RDL w. P. | RDL w. D. | TREELGNN |
|---|---|---|---|---|---|---|
| rel-f1 | **driver-dnf** | $81.49 \pm 0.25$ | $75.19 \pm 2.64$ | $78.31 \pm 0.81$ | $78.64 \pm 0.33$ | $81.90 \pm 0.77$ |
| | **driver-top3** | $89.74 \pm 0.25$ | $76.25 \pm 2.22$ | $77.18 \pm 0.90$ | $79.95 \pm 2.37$ | $89.15 \pm 0.33$ |
| rel-hm | **user-churn** | $70.01 \pm 0.02$ | $69.82 \pm 0.33$ | $70.05 \pm 0.37$ | $69.82 \pm 0.33$ | $69.30 \pm 0.04$ |
| rel-event | **user-ignore** | $91.89 \pm 1.61$ | $90.66 \pm 0.50$ | $90.04 \pm 1.19$ | $92.43 \pm 0.68$ | $91.84 \pm 0.94$ |
| | **user-repeat** | $73.18 \pm 0.44$ | $72.56 \pm 0.79$ | $71.68 \pm 1.32$ | $72.60 \pm 0.90$ | $74.52 \pm 0.46$ |
| rel-stack | **user-engage** | $90.17 \pm 0.03$ | $90.19 \pm 0.05$ | $90.21 \pm 0.03$ | $90.19 \pm 0.05$ | $88.88 \pm 0.02$ |
| | **user-badge** | $87.84 \pm 0.02$ | $89.62 \pm 0.13$ | $89.43 \pm 0.31$ | $89.68 \pm 0.14$ | $84.56 \pm 0.21$ |
| rel-amazon | **user-churn** | $68.79 \pm 0.02$ | $70.45 \pm 0.05$ | $69.89 \pm 0.35$ | $69.91 \pm 0.08$ | $70.01 \pm 0.12$ |
| | **item-churn** | $82.41 \pm 0.02$ | $82.39 \pm 0.02$ | $82.53 \pm 0.16$ | $82.89 \pm 0.11$ | $83.11 \pm 0.04$ |

in the definition of a batch between the two models, we ensured a fair comparison by manually adjusting the batch sizes so that GPU usage was balanced across both models. Table 15 shows the batch sizes and memory consumption for both models.

### F.5 RDL-LESS PARAMETERS

In this section, we compare TREELGNN with a reduced-parameter version of RDL, referred to as RDL small. The number of parameters in RDL small was reduced to match the scale of TREELGNN. Tables 16 and 17 present the metrics (AUCROC/MAE), training time, and inference time. The results demonstrate that reducing the parameters in RDL leads to slightly faster training and inference times but comes at the cost of diminished performance. However, even with this reduction, RDL small remains significantly slower than TREELGNN in both training and inference, highlighting the efficiency advantage of TREELGNN.

## G ABLATION STUDY

We conducted an ablation study to address two key questions: (i) is temporal information necessary for relational database tasks? (ii) Is the distillation of boosted tree models truly essential? To answer these questions, we compare the performance of TREELGNN against two baseline models: TREELGNN W/O TIME, which is a static HeteroGraphSAGE without any temporal information, andTREELGNN W.P., which incorporates temporal information but without distillation, instead directly integrating the row predictions produced by LIGHTGBM as additional node features.

The results provide clear answers to both questions. First, temporal modeling proves to be critical for predictive tasks on relational databases. As shown in Table 18, TREELGNN W/O TIME consistently underperforms when compared to the models that incorporate temporal information (column 1 vs. columns 2 and 3). Second, the distillation process is also essential. TREELGNN significantly

Table 10: Test AUCROC with standard deviation over 5 runs.

| | | LightGBM | RDL | RDL w. P. | RDL w. D. | TreeLGNN |
|---|---|---|---|---|---|---|
| rel-f1 | **driver-dnf** | $70.52 \pm 1.07$ | $71.08 \pm 2.79$ | $69.93 \pm 1.68$ | $71.57 \pm 1.47$ | $73.55 \pm 0.34$ |
| | **driver-top3** | $82.77 \pm 1.08$ | $80.30 \pm 1.85$ | $82.28 \pm 0.76$ | $83.28 \pm 2.47$ | $84.73 \pm 1.43$ |
| rel-hm | **user-churn** | $69.12 \pm 0.01$ | $69.09 \pm 0.35$ | $69.24 \pm 0.56$ | $69.56 \pm 0.35$ | $68.93 \pm 0.03$ |
| rel-event | **user-ignore** | $82.62 \pm 1.14$ | $77.82 \pm 1.88$ | $70.72 \pm 3.70$ | $79.13 \pm 0.60$ | $83.98 \pm 0.44$ |
| | **user-repeat** | $75.78 \pm 1.74$ | $76.50 \pm 0.78$ | $76.57 \pm 1.22$ | $76.63 \pm 1.08$ | $77.77 \pm 0.68$ |
| rel-stack | **user-engage** | $90.34 \pm 0.09$ | $90.59 \pm 0.03$ | $90.66 \pm 0.05$ | $90.50 \pm 0.06$ | $89.02 \pm 0.03$ |
| | **user-badge** | $86.23 \pm 0.04$ | $88.54 \pm 0.15$ | $88.42 \pm 0.29$ | $88.57 \pm 0.22$ | $86.71 \pm 0.54$ |
| rel-amazon | **user-churn** | $68.34 \pm 0.09$ | $70.42 \pm 0.05$ | $69.81 \pm 0.05$ | $69.90 \pm 0.12$ | $69.87 \pm 0.19$ |
| | **item-churn** | $82.62 \pm 0.03$ | $82.82 \pm 0.04$ | $82.93 \pm 0.11$ | $83.12 \pm 0.07$ | $83.84 \pm 0.08$ |

Table 11: Performance of TreeLGNN w/o time with F.E. in AUCROC.

| | rel-f1 driver-top3 | |
|---|---|---|
| | Val. | Test |
| TreeLGNN w/o time | $87.71 \pm 0.51$ | $77.01 \pm 2.44$ |
| TreeLGNN w.P. | $87.38 \pm 0.19$ | $78.77 \pm 0.73$ |
| TreeLGNN | $\mathbf{89.15} \pm 0.33$ | $\mathbf{84.73} \pm 1.43$ |
| TreeLGNN w/o time with F.E. | $87.80 \pm 0.40$ | $78.58 \pm 2.40$ |

outperforms TreeLGNN w.P. (column 2 vs. column 3), demonstrating that embedding the distilled knowledge offers a more informative and effective way to enrich node features than using raw predictions.

Table 12: Comparison of graph construction times for TREELGNN model and RDL.

| | Dataset | Task | TREELGNN | RDL |
|---|---|---|---|---|
| Classification | rel-f1 | driver-dnf | 15 | 7 |
| | | driver-top3 | 17 | 9 |
| | rel-hm | user-churn | 988 | 750 |
| | rel-event | user-ignore | 68 | 407 |
| | | user-repeat | 12 | 54 |
| | rel-stack | user-engagement | 1968 | 1079 |
| | | user-badge | 1342 | 1052 |
| | rel-amazon | user-churn | 558 | 412 |
| | | item-churn | 641 | 486 |
| Regression | rel-f1 | driver-position | 64 | 8 |
| | rel-hm | item-sales | 1818 | 1126 |
| | rel-event | user-attendance | 42 | 68 |
| | rel-stack | post-votes | 1178 | 1033 |
| | rel-amazon | user-ltv | 624 | 486 |
| | | item-ltv | 651 | 501 |

Table 13: End-to-end training time in seconds. TREELGNN model is significantly faster than RDL and performs at a speed comparable to LIGHTGBM.

| | Dataset | Task | TREELGNN | RDL | LIGHTGBM |
|---|---|---|---|---|---|
| Classification | rel-f1 | driver-dnf | 8 | 303 | 4 |
| | | driver-top3 | 5 | 113 | 1 |
| | rel-hm | user-churn | 13 | 3783 | 2 |
| | rel-event | user-ignore | 5 | 96 | 35 |
| | | user-repeat | 3 | 22 | 2 |
| | rel-stack | user-engagement | 745 | 9426 | 217 |
| | | user-badge | 578 | 35791 | 742 |
| | rel-amazon | user-churn | 71 | 3252 | 150 |
| | | item-churn | 87 | 3524 | 174 |
| Regression | rel-f1 | driver-position | 42 | 584 | 32 |
| | rel-hm | item-sales | 83 | 20406 | 1358 |
| | rel-event | user-attendance | 7 | 98 | 14 |
| | rel-stack | post-votes | 728 | 15450 | 387 |
| | rel-amazon | user-ltv | 96 | 2931 | 138 |
| | | item-ltv | 158 | 2991 | 156 |

Table 14: Inference time in seconds. TREELGNN model is significantly faster than RDL and achieves a speed comparable to LIGHTGBM.

|  | Dataset | Task | TREELGNN | RDL | LIGHTGBM |
|---|---|---|---|---|---|
| Classification | rel-f1 | driver-dnf | 0.04 | 0.76 | 0.05 |
|  |  | driver-top3 | 0.05 | 2.19 | 0.04 |
|  | rel-hm | user-churn | 0.01 | 3.63 | 0.30 |
|  | rel-event | user-ignore | 0.02 | 1.19 | 0.02 |
|  |  | user-repeat | 0.01 | 0.29 | 0.04 |
|  | rel-stack | user-engagement | 0.25 | 24.01 | 0.14 |
|  |  | user-badge | 0.25 | 96.23 | 3.03 |
|  | rel-amazon | user-churn | 0.11 | 2.25 | 0.08 |
|  |  | item-churn | 0.09 | 2.24 | 0.08 |
| Regression | rel-f1 | driver-position | 0.07 | 2.34 | 0.04 |
|  | rel-hm | item-sales | 0.01 | 10.52 | 0.13 |
|  | rel-event | user-attendance | 0.01 | 0.95 | 0.04 |
|  | rel-stack | post-votes | 0.23 | 35.23 | 0.79 |
|  | rel-amazon | user-ltv | 0.08 | 5.30 | 0.14 |
|  |  | item-ltv | 0.09 | 5.48 | 0.04 |

Table 15: Batch size comparison: TREELGNN model and RDL are trained using batch sizes that utilize the same amount of memory.

|  | Dataset | Task | TREELGNN | RDL | TREELGNN | RDL |
|---|---|---|---|---|---|---|
|  |  |  | batch size | | Memory usage | |
| Classification | rel-f1 | driver-dnf | All | 64 | 1441 | 1538 |
|  |  | driver-top3 | All | 64 | 1450 | 1508 |
|  | rel-hm | user-churn | All | 32 | 6512 | 6892 |
|  | rel-event | user-ignore | All | 128 | 5364 | 5324 |
|  |  | user-repeat | All | 128 | 5536 | 5622 |
|  | rel-stack | user-engagement | 1 | 256 | 14508 | 15234 |
|  |  | user-badge | 1 | 256 | 14556 | 15058 |
|  | rel-amazon | user-churn | 1 | 1024 | 21554 | 22125 |
|  |  | item-churn | 1 | 1024 | 21589 | 22844 |
| Regression | rel-f1 | driver-position | All | 64 | 1484 | 1548 |
|  | rel-hm | item-sales | All | 32 | 6502 | 6874 |
|  | rel-event | user-attendance | All | 128 | 5466 | 5524 |
|  | rel-stack | post-votes | 1 | 256 | 15122 | 15214 |
|  | rel-amazon | user-ltv | 1 | 1024 | 19326 | 19125 |
|  |  | item-ltv | 1 | 1024 | 19548 | 19584 |

Table 16: RDL with reduced parameters (21,449) for the classification task shows a decrease in performance, with minimal improvements in training and inference time.

| Dataset | Task | AUCROC (↑) | | Training time (seconds) | | Inference time(seconds) | |
|---------|------|-----------|-----------|------------|-----------|-----------|-----------|
| | | RDL small | TREELGNN | RDL small | TREELGNN | RDL small | TREELGNN |
| rel-hm | **user-churn** | 64.28 | **69.56** | 1635 | **13** | 1.39 | **0.01** |

Table 17: RDL with reduced parameters (21,449) for the regression task shows a decrease in performance, with minimal improvements in training and inference time.

| Dataset | Task | MAE (↓) | | Training time (seconds) | | Inference time(seconds) | |
|---------|------|-----------|-----------|------------|-----------|-----------|-----------|
| | | RDL small | TREELGNN | RDL small | TREELGNN | RDL small | TREELGNN |
| rel-hm | **item-sales** | 0.058 | **0.037** | 1909 | **83** | 3.12 | **0.13** |

Table 18: The ablation study proves that (i) temporal modeling is critical for predictive tasks on relational databases and (ii) the distillation process is essential.

| | | | TREELGNN W/O TIME | TREELGNN w.P. | TREELGNN |
|---|---|---|---|---|---|
| Classification | rel-f1 | **driver-dnf** | 68.80 | 70.79 | **73.55** |
| | | **driver-top3** | 77.01 | 78.77 | **84.73** |
| | rel-hm | **user-churn** | 56.07 | **69.38** | 68.93 |
| | rel-event | **user-ignore** | 80.60 | 78.76 | **83.98** |
| | | **user-repeat** | 69.01 | 73.37 | **77.77** |
| | rel-stack | **user-engagement** | 78.58 | **89.94** | 89.02 |
| | | **user-badge** | 81.01 | 84.12 | **86.71** |
| | rel-amazon | **user-churn** | 67.58 | 69.18 | **69.87** |
| | | **item-churn** | 79.58 | 83.37 | **83.84** |
| Regression | rel-f1 | **driver-position** | 5.604 | 3.941 | **3.861** |
| | rel-hm | **item-sales** | 0.055 | 0.038 | **0.037** |
| | rel-event | **user-attendance** | 0.261 | 0.242 | **0.238** |
| | rel-stack | **post-votes** | 0.123 | 0.068 | **0.064** |
| | rel-amazon | **user-ltv** | 16.881 | 14.088 | **13.587** |
| | | **item-ltv** | 57.323 | 49.314 | **48.112** |
| | | *GNN* | ✗ | ✓ | ✓ |
| | | *Time-**then**-graph* | - | ✓ | ✓ |

