# OpenReview forum: "Over 100x Speedup in Relational Deep Learning via Static GNNs and Tabular Distillation"
_ICLR.cc/2025/Conference — Submitted to ICLR 2025_

### Official Review · Reviewer_EvAv · 2024-11-03

**Soundness:** 2
**Presentation:** 2
**Contribution:** 2
**Rating:** 5
**Confidence:** 4

**Summary:**

This paper suggests combining the strengths of tabular models with static Graph Neural Networks (GNNs) to improve performance and speed in predicting key metrics from relational databases. By integrating predictive embeddings from tabular models into a unified GNN framework, the approach achieves up to a 33% performance boost and a remarkable 1050x speedup in inference, making it highly suitable for real-time applications.

**Strengths:**

- **Research Topic.** The studied problem is interesting and practically important for relational deep learning.

- **Writing.** The paper is well-organized and written in a clear, logical manner, moving seamlessly from problem introduction to method presentation, experiments, and results.

**Weaknesses:**

- **Applicability.** The method introduced in this article is limited to application on time series data. Enhancements to enable its application on broader datasets while maintaining substantial time savings would be advantageous.

- **Experimental Baselines.** The paper is suggested to compare experimental results with more baselines, such as XGBoost and static and temporal GNN methods. Only 2 baselines make the experimental section not so convincing.

- **Experimental Datasets.** Only a few part of RelBench is used, and it would be better if more datasets are evaluated on.

- **Experimental Results.** The paper is suggested to compare the time usage with LightGBM, for TREeLGNN can not always outperforms LightGBM.

**Questions:**

- What is the meaning of TREeLGNN?

- In Figure 1, the word "articles" should be "products", and this mistake appears in Section 2.2.

- Figure 2 is not so clear, which needs more explanation.

---

> ### Author Response · Authors · 2024-11-20
>
> We are delighted that the reviewer recognized the practical significance of our research topic and appreciated the clear, logical presentation of our paper. We address the questions in detail below.
>
> >**Q1**: Applicability. The method introduced in this article is limited to application on time series data. Enhancements to enable its application on broader datasets while maintaining substantial time savings would be advantageous.
>
> **A1**: We thank the reviewer for the comment. The focus of our work is to combine pretrained LightGBM with GNNs to efficiently handle temporal e topological information contained in relational databases. Thus, our method is not intended for application to static databases. The time-then-graph framework, which is central to our approach, relies on the temporal structure of the data. We believe that extending the work to static relational databases requires a separate effort we are happy to explore in future work.
>
> >**Q2**: Experimental Baselines. The paper is suggested to compare experimental results with more baselines, such as XGBoost and static and temporal GNN methods. Only 2 baselines make the experimental section not so convincing.
>
> **A2**: To the best of our knowledge, there are only 2 method classes that can be applied to relational databases: GNNs and tree-based methods. We use one representative for each class of methods, following Robinson et al. 2024. Experimenting with other methods within these classes represents an interesting avenue for future research, but one that requires a dedicated effort.
>
> >**Q3**: Experimental Datasets. Only a few part of RelBench is used, and it would be better if more datasets are evaluated on.
>
> **A3**: Our work focuses on node classification and RelBench comprises 7 node-classification datasets and 21 tasks. In our study, we conducted experiments on 4 datasets and 11 tasks. We have expanded our evaluation to include the "Amazon" dataset, encompassing all 4 node-level tasks, thereby increasing our coverage to 5 datasets and 15 tasks.  We did not experiment on the remaining 2 datasets as the authors of RelBench have not yet provided the feature engineering strategy used in LightGBM, which plays a pivotal role in the success of our pipeline.
>
> >**Q4**: Experimental Results. The paper is suggested to compare the time usage with LightGBM, for TREeLGNN can not always outperform LightGBM.
>
> **A4**. We thank the reviewer for the valuable feedback. The focus of our work is to treat pretrained LightGBM as a given from industry, so the time performance of LightGBM is not a primary concern for us. Our goal is to efficiently combine these pretrained models with GNNs to achieve strong performance. Nonetheless, we have now included a table with the end-to-end training and inference times for TREeLGNN, RDL, and LightGBM ( Table 14 in Appendix F.2):
>
> | Dataset| Task| TREeLGNN | RDL| LIGHTGBM |
> |-------------|-----------------|----------|-------|----------|
> | **Classification** |||||
> | rel-f1|driver-dnf|8|303|4|
> | rel-f1|driver-top3|5|113|1|
> | rel-hm|user-churn|13|3783|2|
> | rel-event | user-ignore|5|96|35|
> | rel-event | user-repeat|3|22|2|
> | rel-stack | user-engagement | 745 | 9426  | 217 |
> | rel-stack | user-badge| 578| 35791| 742|
> | rel-amazon | user-churn | 71| 3252| 150|
> | rel-amazon | item-churn | 87| 3524| 174 |
> | **Regression** |||||
> | rel-f1| driver-position | 42 | 584| 32|
> | rel-hm | item-sales | 83 | 20406 | 1358|
> | rel-event| user-attendance | 7 | 98| 14|
> | rel-stack| post-votes | 728 | 15450 | 387|
> | rel-amazon| user-ltv| 96   | 2931  | 138|
> | rel-amazon| item-ltv| 158 | 2991  | 156|
>
> Table 15 in Appendix F.3 shows the inference time in seconds for all the models:
>
> | Dataset | Task | TREeLGNN | RDL | LIGHTGBM |
> |-------------|-----------------|----------|-------|----------|
> | **Classification** |||||
> | rel-f1 | driver-dnf | 0.04 | 0.76 | 0.05 |
> | rel-f1 |driver-top3| 0.05 | 2.19  | 0.04 |
> | rel-hm | user-churn | 0.01| 3.63  | 0.30|
> | rel-event |user-ignore| 0.02| 1.19| 0.02|
> | rel-event |user-repeat| 0.01| 0.29| 0.04|
> | rel-stack|user-engagement | 0.25| 24.01 | 0.14 |
> | rel-stack|user-badge | 0.25| 96.23 | 3.03|
> | rel-amazon | user-churn | 0.11 | 2.25  | 0.08 |
> | rel-amazon| item-churn | 0.09| 2.24 | 0.08|
> | **Regression**  | || | |
> | rel-f1 | driver-position | 0.07| 2.34| 0.04 |
> | rel-hm |item-sales| 0.01| 10.52 | 0.13|
> | rel-event| user-attendance | 0.01| 0.95  | 0.04|
> | rel-stack| post-votes| 0.23| 35.23 | 0.79|
> | rel-amazon| user-ltv| 0.08| 5.30 | 0.16|
> | rel-amazon| item-ltv| 0.09| 5.48 | 0.04|
>
> The training times of TREELGNN are comparable to those of LightGBM. However, we want to highlight that our method achieves an average improvement of 3.9% in regression and 1.4% in classification, resulting in an overall performance gain of 2.42% compared to LightGBM.

---

> ### Author Response · Authors · 2024-11-20
>
> >**Q6**: What is the meaning of TREeLGNN?
>
> **A6**: Thank you for your question, we have added the explanation in the revised manuscript. TREeLGNN stands for:
> - TREe: Referring to the use of tree-based methods,
> - REL: Representing relational deep learning, and
> - GNN: Highlighting our use of graph neural networks.
>
>
>
>
> >**Q7**: In Figure 1, the word "articles" should be "products", and this mistake appears in Section 2.2.
>
> **A7**: Thank you for pointing out this oversight. We have corrected it in Figure 1 and Section 2.2, replacing "articles" with "products."
>
>
> >**Q8**: Figure 2 is not so clear, which needs more explanation.
>
> **A8**: We appreciate your feedback. We have updated the caption for Figure 2 to provide additional explanation and improve the clarity of the figure. In short: “Figure 2. a) For each time $i$, the figure shows the graph $\( \mathcal{G}(V_i) \)$, which represents the interactions occurring at time $i $; b) graph $\( \mathcal{G}(V_{\leq t}) \)$ is the aggregated graph consisting of all interactions from every time step up to $\( t \)$; this is the graph used by RDL, representing the cumulative interactions across all times; c) the graph $\( \mathcal{G}(V_t) \)$ in this panel represents only the interactions that occurred at time $\( t \)$, corresponding to the last graph in panel a), which is the training graph for TREeLGNN.

---

> ### Author Response · Authors · 2024-11-24
>
> Dear Rev EvAv,
> as the discussion period approaches its conclusion, we wanted to ask if you have any additional questions or comments.
>
> We expanded our evaluation to include the "Amazon" dataset and its four tasks, increasing our coverage to 15 tasks across five datasets, utilizing all publicly available feature engineering provided in RelBench. We clarified the meaning of TREeLGNN in the manuscript and corrected terminology issues. Additionally, we improved the clarity of Figure 2 with a more detailed caption, added end-to-end training and inference time comparisons with LightGBM, and highlighted that TREeLGNN achieves an average performance improvement over LightGBM while maintaining comparable training times.

---

> ### Comment · Reviewer_EvAv · 2024-12-03
>
> Thank you very much for your response. I have read it carefully and will maintain my score.

---

### Official Review · Reviewer_bySj · 2024-11-03

**Soundness:** 2
**Presentation:** 2
**Contribution:** 2
**Rating:** 3
**Confidence:** 4

**Summary:**

The paper proposes TREELGNN, which combines LightGBM with GNN.
Basically, the authors consider the graph at time $t$ as a static graph. And then they use the LIGHTGBM-distilled embeddings as node features and use GNN to learn the entity relations. The goal is to predict the result at $t+1$.
The authors conduct experiments on four tasks in RelBench. The performance is on par, but the training and inference time is much faster as compared to RDL.

**Strengths:**

The GNN + LIGHTGBM-distilled embeddings produces much faster training time and inference time, while having slightly better performance on the tasks.

**Weaknesses:**

1. The authors only evaluate their proposed model on one benchmark, RelBench. Moreover, it only compared to 4 out of the 30 tasks from Relbench. It also lacks experiments on link prediction tasks.
2. The idea lacks innovation. RDL already uses deep learning models like resnet for node feature encoder. This paper only changes the encoding strategy from deep learning models to lightgbm. The rest of the architecture follows the design in RDL
3. Using embeddings from pre-trained model will obviously outperform RelBench on speed. This is because the feature encoders in RDL are trained alongside the GNN. So not training the encoder will for sure improve the training/inference time.

**Questions:**

Suggestion:
1. Compare across different tasks in RelBench.
2. The datasets the authors use are fairly small. Consider running experiments on large scale datasets.
3. Consider using other tree-based models than LightGBM.

---

> ### Author Response · Authors · 2024-11-20
>
> We are pleased that the reviewer acknowledged our model's significantly faster training and inference times. They have nonetheless raised a number of important point we wish to clarify in the following.
>
>
> >**Q1**: The authors only evaluate their proposed model on one benchmark, RelBench. Moreover, it only compared to 4 out of the 30 tasks from Relbench. It also lacks experiments on link prediction tasks.
>
>
>
> **A1**: We thank the reviewer for the comment. We only experiment on Relbench since, to the best of our knowledge, it is the only readily available dataset that provides a thorough feature engineering procedure. Moreover, our work focuses on node-level predictions, and RelBench comprises 7 node-level datasets, each associated with multiple predictive tasks, totaling 21 tasks across various domains. In our study, we conducted experiments on 4 datasets and 11 tasks. We have expanded our evaluation to include the "Amazon" dataset, encompassing all 4 node-level tasks, thereby increasing our coverage to 5 datasets and 15 tasks. We could not experiment on the remaining 2 datasets as the feature engineering strategy is not provided in the RelBench code.
>
>
> >**Q2**: The idea lacks innovation. RDL already uses deep learning models like resnet for node feature encoder. This paper only changes the encoding strategy from deep learning models to lightgbm. The rest of the architecture follows the design in RDL
>
> **A2**: Using LightGBM as an encoder allows us to capture temporal interactions, enabling the use of a lightweight static GNN on a graph with reduced neighbourhoods (containing only current-time interactions). In contrast, RDL uses a *temporal aware message passing* GNN on a full graph (appendix A paragraph temporal message passing of [1]), including all interactions, which is computationally more expensive. Furthermore, the choice of LightGBM is motivated by its efficiency in learning meaningful features, including temporal information, which deep learning models like ResNet are not well-suited for in the context of tabular data.
>
> >**Q3**: Using embeddings from pre-trained model will obviously outperform RelBench on speed. This is because the feature encoders in RDL are trained alongside the GNN. So not training the encoder will for sure improve the training/inference time.
>
>
> **A3**: We thank the reviewer for the comment. We believe our performance speedup comes from: 1) the usage of pretrained LGBM encoding, and 2) the employment of a static GNN (which follows from point 1).
>
> >**Q4**: Compare across different tasks in RelBench
>
> **A4**: We would like to clarify that we already compare 11 tasks across 4 datasets. However, we welcome your suggestion, and we have expanded our evaluation to include the additional dataset Amazon along with its corresponding 4 tasks. Regarding the remaining 2 datasets in RelBench, unfortunately, we were unable to include them in our experiments because the feature engineering details required for our method have not been released yet by the authors of RelBench.
>
> >**Q5**: The datasets the authors use are fairly small. Consider running experiments on large scale datasets.
>
> **A5**: In our paper, we have conducted experiments on datasets of substantial scale. Specifically, the 'rel-event' dataset comprises up to 41,000,000 rows, and the 'rel-stack' dataset includes up to 24 relations. These dimensions underscore the robustness of our method in handling large-scale data. Thank you for this comment, we will clarify the size of the datasets in the next revision.
>
> >**Q6**: Consider using other tree-based models than LightGBM.
>
> **A6**: We thank the reviewer for the suggestion.  We would like to highlight that the pipeline we propose is designed to be generic and can indeed accommodate other tree-based models beyond LightGBM. However, exploring the performance variations across different tree-based models falls outside the scope of this work. Our primary goal is not to benchmark different tree-based models but to introduce a new framework tailored to pretrained temporal relational database models that focus more on the temporal component of the predictions than the relational one (such as boosted decision trees). We chose LightGBM as it was the model provided by the RelBench paper, which contains extensive feature engineering
>
> [1] Fey, Matthias, et al. "Position: Relational Deep Learning-Graph Representation Learning on Relational Databases." Forty-first International Conference on Machine Learning.

---

> > ### Comment · Reviewer_bySj · 2024-11-25
> > **Authors understanding of RDL is wrong.**
> >
> > Thank you for the response. I am happy to see the new experiments. However, I think the answer to Q2 is not satisfactory. I disagree that RDL uses full graph or RDL uses temporal GNNs. In fact, RDL samples subgraphs and it uses a static GNNs snapshotted at different seed times.

---

> ### Author Response · Authors · 2024-11-24
>
> Dear Rev bySj,
> as the discussion period approaches its conclusion, we wanted to ask if you have any additional questions or comments.
>
> In short, we expanded our evaluation by adding the "Amazon" dataset and its four tasks, increasing our coverage to a total of 15 tasks. Our experiments encompass **all tasks** available in the RelBench benchmark repository that include publicly provided feature engineering. We clarified the size and scale of the datasets used. Finally, we highlighted the generic nature of our framework.

---

> ### Author Response · Authors · 2024-11-26
>
> Thank you for the response.
>
> > I disagree that RDL uses full graph or RDL uses temporal GNNs.
>
>
> As detailed in [1], RDL introduces temporal message passing (Equation 5, Appendix A), where messages are propagated exclusively to neighbors from preceding time steps. To enhance scalability, RDL further employs a sampling strategy across all neighbors (Section 3.3 of [1]). Notably, the definition of a temporal GNN is inherently complex, and RDL explicitly frames their approach under the concept of "temporal message passing" (Appendix A of [1]), and thus, this is how they handle the time dimension. We use the term temporal GNN to refer to a temporal-aware message-passing approach.
>
> In fact, we explicitly address temporal dynamics by leveraging the capabilities of LightGBM. By allowing LightGBM to manage temporal information explicitly, we go beyond the limitations of RDL's implicit temporal handling, providing a more effective mechanism for capturing temporal dependencies.
>
>
> We thank you for the opportunity to further clarify these points and we remain available for any further question.
>
>
> [1] Fey, Matthias, et al. "Position: Relational Deep Learning-Graph Representation Learning on Relational Databases." Forty-first International Conference on Machine Learning.

---

> ### Comment · Reviewer_bySj · 2024-11-27
>
> Thank you for your reply. However, RDL does not use temporal graphs, all the edges and nodes in the graphs are static. Timestamps are passed as features. I agree that it uses temporal sampling strategy but I think there's a misunderstanding on what is considered as temporal GNNs.
>
> Second, RDL can sample all neighbors, but that only limits to the subgraph, not the entire graph. So if the algorithm samples a 2-hop subgraph, $user_1$ -> $trans_1$ -> $item_1$ and $item_1$ is an isolated node, then in this case how is the entire graph sampled?

---

> > ### Author Response · Authors · 2024-11-28
> >
> > Thank you for your response.
> >
> > > However, RDL does not use temporal graphs, all the edges and nodes in the graphs are static. Timestamps are passed as features. I agree that it uses temporal sampling strategy but I think there's a misunderstanding on what is considered as temporal GNNs.
> >
> > However, regardless of whether a static GNN with temporal-aware message passing is classified as a temporal-GNN (which is beyond the scope of this paper), the novelty of our method lies in its ad-hoc mechanism for capturing temporal dependencies, combined with the use of a GNN on an enormously smaller static network. This approach enables our model to achieve an average performance improvement of 2.42% while significantly reducing computational time compared to RDL.
> >
> > > Second, RDL can sample all neighbors, but that only limits to the subgraph, not the entire graph. So if the algorithm samples a 2-hop subgraph, user1 -> trans1   -> item1   and item1 is an isolated node, then in this case how is the entire graph sampled?
> >
> >
> > In the case you mentioned, if a neighbour sampler samples **all** the neighbours, and the resulting subgraph is (user1 → trans1 → item1). it means that it is a disconnected component in the original graph. Generally, If you sample **all** neighbours for every node in the graph, you will effectively reconstruct the entire graph.
> >
> > On the other hand, when the number of hops is limited (e.g., using 2-hop convolutions), the GNN will only aggregate information within that limited scope. For example, in a graph like user1 → trans1 → item1 → store1, a 2-hop convolution will not incorporate information from "store1" into the representation of "user1" because it is beyond the 2-hop radius. This is a standard characteristic of GNNs—they aggregate information only from the specified number of hops.
> >
> > By “the entire graph,” we refer to the portion of the graph that the GNN is capable of sampling based on the hop constraints. In this sense, RDL ensures that the number of hops in the sampling algorithm matches the number of layers in the GNN, which is how the graph is effectively sampled and processed.
> >
> > For reference, here is a piece of code taken from the official rel-bench repository that shows how the sampling algorithm works:
> > ```
> > loader_dict[split] = NeighborLoader(
> >         data,
> >         num_neighbors=[int(args.num_neighbors / 2**i) for i in range(args.num_layers)],
> >         time_attr="time",
> >         input_nodes=table_input.nodes,
> >         input_time=table_input.time,
> >         transform=table_input.transform,
> >         batch_size=args.batch_size,
> >         temporal_strategy=args.temporal_strategy,
> >         shuffle=split == "train",
> >         num_workers=args.num_workers,
> >         persistent_workers=args.num_workers > 0,
> >     )
> > ```
> > To clarify, complex combinations involving a mismatch between the number of layers and the number of hops in the sampling algorithm are not considered in either RDL or in our approach.
> >
> >
> > We would like to thank the reviewer once again for providing valuable comments and engaging in the discussion.
> > We deeply appreciate it.

---

> ### Comment · Reviewer_bySj · 2024-11-29
>
> But given the implementation shown in your response, RDL does not sample the full graph in each mini-batch, it only samples $batch\_size$ number of nodes. In fact, $batch_size$ is set between 128 to 512. Moreover, RDL provides a `num_neighbors` argument, which you can specify on how large the subgraph size can be. This means sampling the full graph is not a prerequisite or constraint for running RDL.

---

> > ### Author Response · Authors · 2024-11-29
> >
> > RDL uses neighborhood sampling at the implementation level for scalability reason. However, we would like to emphasize that this is fundamentally distinct from the model-level definition.
> >
> > In RDL, the model is defined over the entire graph, and the neighborhood sampling is an implementation choice to manage computational complexity. In contrast, our approach defines a different training graph at the model level. This training graph is structurally smaller and tailored to the task but is not merely a subset obtained through neighborhood sampling. Instead, it represents a distinct design choice.
> >
> > This distinction between model definition and implementation optimization highlights a key difference in our methodological approach.

---

### Official Review · Reviewer_qwSi · 2024-11-04

**Soundness:** 2
**Presentation:** 1
**Contribution:** 2
**Rating:** 3
**Confidence:** 5

**Summary:**

This paper proposes to combine the strengths of tabular models and static GNNs to achieve more efficient predictions on relational databases. The reported predictive and runtime performance are quite significant.

**Strengths:**

- Strong predictive performance and very strong reported runtime efficiency
- The proposed method is intuitive and easy to understand

**Weaknesses:**

- The proposed idea is not novel. There is a long history of combining tree-based models (e.g., XGBoost) with GNNs, including using XGBoost as features for GNNs as is used in this paper; however, the paper fails to discuss this line of related works. For example, Boost then Convolve: Gradient Boosting Meets Graph Neural Networks, accepted to ICLR 2021, also proposed similar ideas, but the paper failed to mention it.
- The premise of this paper, that TGNNs are mainly trained over the same relational data, is wrong/questionable. If we check the related works cited in this paper, we can find that many existing works do not use a TGNN but use the static GNN that is also used in this paper. Even for the paper that this paper frequently refers to and compares against (Robinson et al.,2024), they did not claim they are using a TGNN either.
- The major claim, that there is a 100x speed-up, is questionable. To make convincing claims on runtime efficiency boost, the paper should have defined the relevant settings and metrics clearly; however, the paper failed to do so. Some of the questions/concerns include: (1) Does the tree-based model (LightGBM) component count as the training/inference time? (2) If so, why not report the training and inference time of LightGBM? Its predictive performance has been reported, but its runtime is missing. (3) What is the definition of epoch here? The mini-batch definition of a temporal GNN and a static GNN could be quite different. (4) Instead of reporting runtime per epoch which is unclear, why not report the complete end-to-end training/inference time for all the methods? (5) Based on the appendix (by the way, according to ICLR guidelines, the appendix should be attached with the submission, but this paper apparently did not follow the guideline), the number of parameters of temporal GNN is orders of magnitude large than static GNN. For example, for rel-hm the RDL has 100x more parameters than the proposed method, which is not a fair comparison for arguments on runtime.
- Following the above-mentioned concerns, explicitly claiming the over 100x speedup in the title is not appropriate and could be misleading.
- The implementation of the baseline is also very vague. After checking, the RDL was originally a position paper. There are many variants of RDL, including GraphSAGE and ID-GNN versions. The paper does not mention which version of RDL they use in all the comparisons. Moreover, all the numbers in this paper are not the same as the reported number of RDL and the baseline LightGBM. The authors could better explain how the numbers are obtained.

**Questions:**

- Across all the experiments, it seems that the predictive performance between LightGBM and proposed TreeLGNN is quite marginal, in some cases, such as rel-hm or rel-stack, LightGBM is actually better. Is GNN really helpful for the RDB prediction tasks if efficiency is the major focus? Could the author report the inference/training speed up from LightGBM and over the proposed TreeLGNN?

- There are many relational databases that are not dynamic, why does this work focus on the temporal aspect of RDB prediction tasks? For static RDB prediction tasks, I suppose the proposed method will actually be slower. Here, the baseline will also be a vanilla static GNN, and the proposed TreeLGNN will have an additional LightGBM preprocessing module.

---

> ### Author Response · Authors · 2024-11-20
>
> We are pleased to see that the reviewer recognized the strong predictive performance and the runtime efficiency of our method, as well as its intuitive and easy-to-understand approach.
>
>
> >**Q1:** The proposed idea is not novel. There is a long history of combining tree-based models (e.g., XGBoost) with GNNs, including using XGBoost as features for GNNs as is used in this paper.
>
> **A1**: We thank the reviewer for their feedback. We would like to clarify that, despite being related, these works differ from ours in the main motivation, as they focus on enhancing standard GNNs on graph data which is not from a relational database (therefore, without temporality nor heterogeneity). On the contrary, our goal is to leverage pretrained tree-based models (LIGHTGBM) to capture the temporal component of a temporal graph model, aiming to speed up relational deep learning models. Moreover, to the best of our knowledge, existing methods tend to replace trees with GNNs in a boosting setup [1-5]. Instead, our work aims at employing pre-trained tree-based methods as a component of our framework, specifically to capture the temporality of our data, and then integrate that with a static GNN capturing instead the relationships within our data.
>
> We will add to the related work section a paragraph to better distinguish our approach from boosted GNN methods such as [1-5] and explaining the differences with respect to our TREeLGNN.
>
> [1] Sun et al. AdaGCN: Adaboosting Graph Convolutional Networks into Deep Models. ICLR 2021.
>
> [2] Shi et al. Boosting-GNN: Boosting Algorithm for Graph Networks on Imbalanced Node Classification. Frontiers in Neurorobotics 2021.
>
> [3] Zheng et al. AdaBoosting Clusters on Graph Neural Networks. IEEE CDM 2021.
>
> [4] D Tang et al. XGNN: Boosting Multi-GPU GNN Training via Global GNN Memory Store. VLDB 2024.
>
> [5] D Deng et al. "XGraphBoost: extracting graph neural network-based features for a better prediction of molecular properties." Journal of chemical information and modeling 2021.
>
> [6] Ivanov et al. Boost then Convolve: Gradient Boosting Meets Graph Neural Networks. ICLR 2021.
>
> >**Q2:** many existing works do not use a TGNN but use the static GNN that is also used in this paper. Even for the paper that this paper frequently refers to and compares against (Robinson et al.,2024), they did not claim they are using a TGNN either.
>
> **A2:** We thank the reviewer for the opportunity to clarify an important point. While the authors of RDL do not explicitly refer to their approach as a TGNN (saying they use a "GNN with temporal awareness."), *RDL is a TGNN*. Specifically, their message passing with temporal awareness is defined in equations (4) and (5) in Appendix A of (Fey et al.,2024), which is a Temporal GNN message-passing procedure. Moreover, we would like to clarify that our paper does not simply propose to replace a TGNN with a static GNN. On the contrary, our paper argues in favor of employing pre-trained tree-based methods, which excels at capturing the temporality of the data also due to extensive feature engineering. Then, since the temporality is captured by the tree-based method, we can employ a static GNN rather than a TGNN.
>
>
> >**Q3:**  Does the tree-based model (LightGBM) component count as the training/inference time?
>
> **A3:** The reported training and inference times do not include the LightGBM component, which is treated as preprocessing.
> LightGBM is trained once at the beginning, separate from the end-to-end model training. Nonetheless, we welcome your suggestion, and we have added the preprocessing times to the Appendix F, including Table 13, which details the total preprocessing time for both TREeLGNN and RDL. Specifically, to construct the graphs used by TREeLGNN, LightGBM predictions must first be computed and distilled. In contrast, building the graph for RDL requires constructing the temporal graph and embedding the input features.
>
> | Dataset     | Task            | TREeLGNN | RDL  |
> |-------------|-----------------|----------|------|
> | **Classification** | | | |
> | rel-f1      | driver-dnf  | 15       | 7    |
> |             | driver-top3 | 17       | 9    |
> | rel-hm      | user-churn  | 988      | 750  |
> | rel-event   | user-ignore  | 68       | 407  |
> |             | user-repeat     | 12       | 54   |
> | rel-stack   | user-engagement | 1968     | 1079 |
> |             | user-badge      | 1342     | 1052 |
> | rel-amazon  | user-churn      | 558      | 412  |
> |             | item-churn      | 641      | 486  |
> | **Regression**   | | | |
> | rel-f1      | driver-position | 64       |    8  |
> | rel-hm      | item-sales      | 1818     | 1126 |
> | rel-event   | user-attendance | 42       | 68   |
> | rel-stack   | post-votes      | 1178     |  1033    |
> | rel-amazon  | user-ltv  | 624      | 486  |
> |         | item-ltv        | 651      | 501  |
>
> TREeLGNN’s preprocessing is sometimes double that of RDL but is offset by significantly faster training, leading to a more efficient overall pipeline.

---

> ### Author Response · Authors · 2024-11-20
>
> >**Q4:** If so, why not report the training and inference time of LightGBM? Its predictive performance has been reported, but its runtime is missing.
>
> **A4:** We thank the reviewer for this valuable suggestion. We have now included the end-to-end training and inference times for TREeLGNN, RDL, and LightGBM. Table 14 in Appendix F.2 shows the end-to-end training time in seconds for all the models:
>
> | Dataset     | Task         | TREeLGNN | RDL   | LIGHTGBM |
> |-------------|-----------------|----------|-------|----------|
> | **Classification** |                 |          |       |          |
> | rel-f1      | driver-dnf      | 8        | 303   | 4        |
> |             | driver-top3     | 5        | 113   | 1        |
> | rel-hm      | user-churn      | 13       | 3783  | 2        |
> | rel-event   | user-ignore     | 5        | 96    | 35       |
> |             | user-repeat     | 3        | 22    | 2        |
> | rel-stack   | user-engagement | 745      | 9426  | 217      |
> |             | user-badge      | 578      | 35791 | 742      |
> | rel-amazon  | user-churn      | 71       | 3252  | 150      |
> |             | item-churn      | 87       | 3524  | 174      |
> | **Regression**   |                 |          |       |          |
> | rel-f1      | driver-position | 42       | 584   | 32       |
> | rel-hm      | item-sales      | 83       | 20406 | 1358     |
> | rel-event   | user-attendance | 7        | 98    | 14       |
> | rel-stack   | post-votes      | 728      | 15450 | 387      |
> | rel-amazon  | user-ltv        | 96       | 2931  | 138      |
> |             | item-ltv        | 158      | 2991  | 156      |
>
> Table 15 in Appendix F.3 shows the inference time in seconds for all the models:
>
> | Dataset     | Task            | TREeLGNN | RDL   | LIGHTGBM |
> |-------------|-----------------|----------|-------|----------|
> | **Classification** |                 |          |       |          |
> | rel-f1      | driver-dnf      | 0.04     | 0.76  | 0.05     |
> |             | driver-top3     | 0.05     | 2.19  | 0.04     |
> | rel-hm      | user-churn      | 0.01     | 3.63  | 0.30     |
> | rel-event   | user-ignore     | 0.02     | 1.19  | 0.02     |
> |             | user-repeat     | 0.01     | 0.29  | 0.04     |
> | rel-stack   | user-engagement | 0.25     | 24.01 | 0.14     |
> |             | user-badge      | 0.25     | 96.23 | 3.03     |
> | rel-amazon  | user-churn      | 0.11     | 2.25  | 0.08     |
> |             | item-churn      | 0.09     | 2.24  | 0.08     |
> | **Regression**   |                 |          |       |          |
> | rel-f1      | driver-position | 0.07     | 2.34  | 0.04     |
> | rel-hm      | item-sales      | 0.01     | 10.52 | 0.13     |
> | rel-event   | user-attendance | 0.01     | 0.95  | 0.04     |
> | rel-stack   | post-votes      | 0.23     | 35.23 | 0.79     |
> | rel-amazon  | user-ltv        | 0.08     | 5.30  | 0.16     |
> |             | item-ltv        | 0.09     | 5.48  | 0.04     |
>
> The training times are comparable between TREeLGNN and LightGBM. However, we would like to emphasize that our work is based on the idea of having LightGBM pre-trained and readily available, which is common practice in industry.

---

> ### Author Response · Authors · 2024-11-20
>
> >**Q5:** What is the definition of epoch here? The mini-batch definition of a temporal GNN and a static GNN could be quite different.
>
> **A5:** We thank the reviewer for highlighting this important point. In the case of TREeLGNN, a batch of size 1 corresponds to the entire graph at the current timestamp, as the model processes the full set of interactions at that time. In contrast, for RDL, a batch of size 1 refers to a single node along with its temporal neighborhood. Given these differences in the definition of a batch between the two models, we ensured a fair comparison by manually adjusting the batch sizes so that GPU usage was balanced across both models. In Appendix F.4, Table 16, we provide the batch size used and the memory consumption for both RDL and TREeLGNN.
> | Dataset       | Task            | batch size | Memory usage | batch size | Memory usage |
> |---------------|-----------------|------------|--------------|------------|--------------|
> |               |                 | TREeLGNN   | RDL          | TREeLGNN   | RDL          |
> | Classification|                 |            |              |            |              |
> | rel-f1        | driver-dnf      | All        | 64           | 1441       | 1538         |
> |               | driver-top3      | All        | 64           | 1450       |   1508           |
> | rel-hm        | user-churn      | All        | 32           | 6512       | 6892         |
> | rel-event     | user-ignore     | All        | 128          | 5364       | 5324         |
> |               | user-repeat     | All        | 128          | 5536       | 5622         |
> | rel-stack     | user-engagement | 1          | 256          | 14508      | 15234        |
> |               | user-badge      | 1          | 256          | 14556      | 15058        |
> | rel-amazon    | user-churn      | 1          | 1024         | 21554      | 22155        |
> |               | item-churn      | 1          | 1024         | 21589      | 22484        |
> | Regression    |                 |            |              |            |              |
> | rel-f1        | driver-position | All        | 64           | 1484       | 1548         |
> | rel-hm        | item-sales      | All        | 32           | 6502       | 6874         |
> | rel-event     | user-attendance | All        | 128          | 5466       | 5524         |
> | rel-stack     | post-votes      | 1          | 256          | 15122      | 15124        |
> | rel-amazon    | user-ltv        | 1          | 1024         | 19326      | 19125        |
> |               | item-ltv        | 1          | 1024         | 19548      | 19584        |
>
> >**Q6:** Instead of reporting runtime per epoch which is unclear, why not report the complete end-to-end training/inference time for all the methods?
>
> **A6:** In response to the reviewer's suggestion, we have included the tables presented in Q4.
>
> >**Q7**: Based on the appendix (by the way, according to ICLR guidelines, the appendix should be attached with the submission, but this paper apparently did not follow the guideline), the number of parameters of temporal GNN is orders of magnitude large than static GNN. For example, for rel-hm the RDL has 100x more parameters than the proposed method, which is not a fair comparison for arguments on runtime.
>
> **A7**: We thank the reviewer for the comment. We view this as a key strength of our method—despite using fewer parameters, TREeLGNN achieves better or comparable performance. However, to provide a more thorough comparison, we have added supplementary material in Appendix F.5 (Table 17 and Table 18), where we show that reducing the number of parameters in RDL to match TREeLGNN's parameter count significantly reduces its performance. Nevertheless, TREeLGNN still maintains its advantage in runtime efficiency, reinforcing its ability to perform faster while achieving competitive results.
> We apologize for not including the Appendix in the initial submission; we have corrected this oversight. Thank you!
>
>
> | Dataset   | Task       |  model | AUROC (↑) | Training time (seconds) | Inference time (seconds) |
> |-----------|------------|----|-------|-------------------------|--------------------------|
> | rel-hm    | user-churn | RDL small | 64.28      | 1635    | 1.39 |
> |           |            | TREeLGNN  | 69.56                   | 13       | 0.01 |
>
>
> | Dataset   | Task | model       | MAE (↓) | Training time (seconds) | Inference time (seconds) |
> |-----------|--------|-----|---------|-------------------------|--------------------------|
> | rel-hm    | item-sales  |RDL small | 0.058   | 1909 | 3.12 | 0.13 |
> | | | TREeLGNN  | 0.037  | 83 |  0.13 |

---

> ### Author Response · Authors · 2024-11-20
>
> >**Q8**: Following the above-mentioned concerns, explicitly claiming the over 100x speedup in the title is not appropriate and could be misleading.
>
> **A8**: We appreciate the reviewer’s feedback. We would like to emphasize that the time required by LightGBM (or similar models) has not been considered, as we assume such tabular models are readily available and provided by industry. However, we understand the concern and we propose revising the title to:
> “Combining Pretrained Tabular Models with Static GNNs in Relational Deep Learning”
>
>
> >**Q9**: There are many variants of RDL, including GraphSAGE and ID-GNN versions. The paper does not mention which version of RDL they use in all the comparisons.
>
> **A9**: In our work we concentrate on node level predictions. For node-level tasks, the RDL paper uses the GraphSAGE variant, which is the version we applied in our comparisons; the ID-GNN variant of RDL is instead used exclusively for link prediction tasks. We thank you for this comment and we will clarify in the next revision.
>
>
> >**Q10**: All the numbers in this paper are not the same as the reported number of RDL and the baseline LightGBM. The authors could better explain how the numbers are obtained.
>
> **A10**: We re-ran all the experiments to ensure fairness in the comparison. We remark however that the differences between the numbers reported in the RDL paper and the ones in our tables are within the standard deviation. We will make this point clearer, thank you.
>
>
> >**Q11**: Across all the experiments, it seems that the predictive performance between LightGBM and proposed TreeLGNN is quite marginal, in some cases, such as rel-hm or rel-stack, LightGBM is actually better. Is GNN really helpful for the RDB prediction tasks if efficiency is the major focus? Could the author report the inference/training speed up from LightGBM and over the proposed TreeLGNN?
>
> **A11**: While efficiency represents an important aspect of our goal, we do not view it as the major focus, instead we assume we have access to a LightGBM pretrained model, which we aim to use resulting in both improved performance and faster runtime. Nonetheless, we have included additional results in Appendix F.2 and Appendix F.3 (Table 14 and Table 15 which we have reported in here Q4), which compare the inference/training speed of LightGBM versus TREeLGNN. While LightGBM is up to 7 times faster in some cases, it is important to note that TREeLGNN outperforms LightGBM in 13 out of 15 tasks (considering also the dataset Amazon that we have now included). Specifically, TREeLGNN shows an average improvement of 3.9% in regression and 1.4% in classification, resulting in a total performance improvement of 2.42%.
>
>
> >**Q12**: There are many relational databases that are not dynamic, why does this work focus on the temporal aspect of RDB prediction tasks? For static RDB prediction tasks, I suppose the proposed method will actually be slower. Here, the baseline will also be a vanilla static GNN, and the proposed TreeLGNN will have an additional LightGBM preprocessing module.
>
> **A12**:  We thank the reviewer for the comment. Our goal is to employ a pretrained LightGBM model to encode the temporal dynamics of the data so TREeLGNN is specifically designed for temporal relational databases and is not intended for use on static databases. The key component of our approach, the time-then-graph framework, relies on the temporal aspect of the data, making it less suitable for static databases where this structure does not apply. We believe that exploring how TREeLGNN can be adapted to static relational databases represents an interesting avenue for future research.

---

> ### Author Response · Authors · 2024-11-24
>
> Dear Rev qwSi,
> as the discussion period approaches its conclusion, we wanted to ask if you have any additional questions or comments.
>
>
> In short, we clarified the novelty of TREeLGNN by distinguishing it from related methods and explaining its specific focus on temporal relational databases. We expanded the related work section, added preprocessing times, and reported end-to-end training and inference times, including detailed parameter comparisons and runtime analyses. Furthermore, we show that TREeLGNN outperforms RDL even when RDL's parameters are reduced, in both performance and inference/training time. Additionally, we revised the title to avoid misleading claims, and included missing appendices.

---

### Official Review · Reviewer_QhxJ · 2024-11-04

**Soundness:** 2
**Presentation:** 2
**Contribution:** 3
**Rating:** 5
**Confidence:** 4

**Summary:**

This paper introduces TREELGNN, a novel method for addressing relational database (RDB) prediction. TREELGNN combines advanced feature engineering from tabular models with a more efficient static GNN backbone. This approach demonstrates improved performance and computational efficiency compared to the Temporal GNN method when evaluated on RelBench.

**Strengths:**

1. The paper presents a straightforward yet effective method that shows significant efficiency improvements over Temporal GNNs for RDB prediction tasks.
2. The authors’ approach to modeling tabular features, RDB relationships, and temporal dynamics offers a unique perspective. The framework’s emphasis on feature modeling, while simplifying temporal dynamics, proves effective.
3. The framework is flexible and extendable, supporting a wide selection of tabular models and GNN layers, allowing for more potential configurations.

**Weaknesses:**

1. The effectiveness of the proposed distillation process is unclear. Distillation generally involves training an additional encoder to transfer information from the tabular model to the main predictive model. However, the analysis does not clarify why this method outperforms directly incorporating feature-engineered results from the tabular models. (See Question Q1 for more on this.)
2. The distinction between the “time-then-graph” process in TREELGNN and previous TGNN methods is insufficiently explained. The authors mention time-then-graph as a computation-saving approximation for temporal GNNs. However, this concept has already been established in temporal GNN literature (also as the authors cited), as seen with models like TGN [1], which uses memory to store previous information while processing only current events—closely resembling TREELGNN. Further comparison would help clarify TREELGNN’s unique contributions in this regard.
3. The paper contains minor typographical errors, such as the misuse of commas and decimal points in tables.

[1] Temporal graph networks for deep learning on dynamic graphs https://arxiv.org/abs/2006.10637

**Questions:**

1. Could the authors more systematically compare the candidate baselines, such as LIGHTGBM, RDL w.P, RDL w.D, and TREELGNN w.P? Specifically:
- (a) Why might RDL with LIGHTGBM predictions perform worse than LIGHTGBM alone (sometimes markedly worse)?
- (b) What would be the effect of distilling information using the GNN outputs? For instance, if embeddings from the original TGNN or TREELGNN w.P were used to optimize final predictions with LIGHTGBM outputs?
- (c) Performance improvements in classification appear inconsistent across RDL w.P, RDL w.D, and TREELGNN. What might be contributing to this variability?
- (d) Appendix C provides additional experiments on distillation; could the authors elaborate on these experiments and their conclusions?
2. What accounts for the substantial efficiency gains observed? Are they due to TGNNs' inefficient temporal neighbor accesses, or simply fewer neighbors in general? Would restricting TGNN to a similar number of neighbors reduce its computational time as well?
3. A suggestion for presenting results: A pipeline diagram might help provide a clearer overview of the approach. Additionally, organizing the experiment results with a clearer separation between main results and ablation studies might improve clarity, as the current presentation mixes the two.

---

> ### Author Response · Authors · 2024-11-20
>
> We are delighted to see the reviewer appreciating the efficiency improvements of our TREeLGNN, as well as the unique approach to modeling tabular features, relationships, and temporal dynamics. The reviewer asked a number of important questions we clarify in the following.
>
> >**Q1:** The effectiveness of the proposed distillation process is unclear. Distillation generally involves training an additional encoder to transfer information from the tabular model to the main predictive model. However, the analysis does not clarify why this method outperforms directly incorporating feature-engineered results from the tabular models.
>
> **A1:** That is a great question! Thank you for your insight. We believe the effectiveness of the proposed distillation process arises from how it allows the model to extract more information from a hard-label predictor by softening its outputs, as shown in (Hinton et al. 2015). Empirically, this is also supported by our experiments, where RDL w.P. performs worse than RDL w.D., specifically in 12 out of 15 tasks. This demonstrates that distillation helps capture richer information, improving performance over using hard labels directly.
> [1] G. Hinton, O. Vinyals, J. Dean, Distilling the knowledge in a neural network, arXiv:1503.02531, 2015.
>
> >**Q2**: why might RDL with LIGHTGBM predictions perform worse than LIGHTGBM alone (sometimes markedly worse)?
>
> **A2:** Thank you for the observation. We would like to point out that there are only 4 out of 15 cases where RDL + LIGHTGBM does not outperform LIGHTGBM alone. In 3 out of these 4 cases, the difference in performance remains within the standard deviation. Therefore, only in 1 case RDL + LIGHTGBM performs significantly worse than LIGHTGBM alone, namely on the rel-event dataset and user-ignore task.
> We speculate that, in this case, the RDL model is unable to effectively utilize the prediction. This may be because the node features in the user-ignore task are quite large, and adding just a single value for the prediction is insufficient hint for the model to understand its importance.
>
> >**Q3:** What would be the effect of distilling information using the GNN outputs? For instance, if embeddings from the original TGNN or TREELGNN w.P were used to optimize final predictions with LIGHTGBM outputs?
>
> **A3:** That is an interesting question. If we understood correctly, the reviewer is asking “what if TGNN or TREeLGNN w.P were pretrained and we would distill them to use as inputs to LIGHTGBM?”. We feel this research direction is out of our main scope, since our work focuses on using existing pretrained LIGHTGBM models (these models are widely deployed in industry), rather than pretrained GNN-type models.
>
> >**Q4:** Performance improvements in classification appear inconsistent across RDL w.P, RDL w.D, and TREeLGNN. What might be contributing to this variability.
>
> **A4**: We believe the reason for this inconsistency is due to these methods being good at capturing different types of relational-temporal information from the data. That is, if a given task requires learning a specific type of relation-temporal pattern, then a specific method that captures that type of information well could outperform others. Because our method encompasses two distinct ways to capture temporal-relational patterns, it should be more consistent than other methods, outperforming both RDL w.P and RDL w.D in 11 tasks out of 15. Whether there exists a model that can capture all such patterns remains an open problem.
>
> >**Q5:** Appendix C provides additional experiments on distillation; could the authors elaborate on these experiments and their conclusions?
>
> **A5:** We thank the reviewer for the opportunity to expand the discussion on our experiments. We have expanded the discussion in Appendix C. For your convenience, here is a short summary:
>
> “We perform a knowledge distillation from LIGHTGBM into an MLP; we report AUC-ROC/MAE against the true target, and against the predictions of LIGHTGBM. The distillation results show that the process has been effectively carried out: the MAE and AUCROC with respect to the targets predicted by LightGBM are on average x and y, respectively, demonstrating that the knowledge has been successfully transferred. However, in general, compared to the real target, the performance is slightly lower than LightGBM, with values being on average z lower than those from LightGBM. This highlights that, although knowledge is distilled, LightGBM's ability to perform well on tabular data remains unmatched”

---

> > ### Author Response · Authors · 2024-11-20
> >
> > >**Q6**: The distinction between the “time-then-graph” process in TREELGNN and previous TGNN methods is insufficiently explained. The authors mention time-then-graph as a computation-saving approximation for temporal GNNs. However, this concept has already been established in temporal GNN literature (also as the authors cited), as seen with models like TGN [1], which uses memory to store previous information while processing only current events—closely resembling TREELGNN. Further comparison would help clarify TREELGNN’s unique contributions in this regard.
> >
> > **A6**: Thanks for the opportunity to clarify our contributions. Indeed, there are multiple “time-then-graph” methods. Our contribution is to have as a temporal component of the “time-then-graph” framework a boosted decision tree, since these generally heavily use temporal features. Moreover, we believe that adapting TGN to relational databases, despite being possible, is non-trivial due to the ad hoc encoding of features. We believe properly studying the performance of TGN in relational databases falls outside of the scope of the current paper, which instead compares to the TGNN proposed in RelBench.
> >
> > >**Q7:** What accounts for the substantial efficiency gains observed? Are they due to TGNNs' inefficient temporal neighbor accesses, or simply fewer neighbors in general? Would restricting TGNN to a similar number of neighbors reduce its computational time as well?
> >
> > **A7:** We thank the reviewer for the comment. The substantial efficiency gains observed in TREeLGNN are due to two main factors: 1) the efficiency of LightGBM compared to ResNet for end-to-end training, and 2) the reduced complexity of the training graph in TREeLGNN.
> > First, LIGHTGBM computational cost is mostly on pre-processing (which can be very efficient even in large databases with SQL queries on large-scale commercial systems). Since we consider the LIGHTGBM to be pretrained (as they are already deployed in industry), our TREeLGNN takes advantage of this pretraining. Second, since we consider LIGHTGBM as capturing most of the temporal information, TREeLGNN's graph is a static graph capturing the relation among entities.
> > For these reasons, restricting the number of neighbors would not substantially reduce the computational time in RDL. While fewer neighbors could help reduce some of the complexity, the primary issue lies in the inefficient feature encoding process. The ResNet-based encoding of tabular features is heavy, as it is not well-suited for handling relational data. Removing neighbors without addressing the temporal encoding in another way would also result in a loss of temporal information, which is a critical aspect of the model's effectiveness.
> >
> > >**Q8**: A suggestion for presenting results: A pipeline diagram might help provide a clearer overview of the approach. Additionally, organizing the experiment results with a clearer separation between main results and ablation studies might improve clarity, as the current presentation mixes the two.
> >
> > **A8:** That is a really great suggestion! We have added a diagram in the main text to illustrate the pipeline and have reorganized the experimental section, clearly separating the results from the ablation studies.
> >
> > >**Q9:** The paper contains minor typographical errors, such as the misuse of commas and decimal points in tables.
> >
> > **A9:** Thank you for your careful reading of our paper, we have fixed them in our revision.

---

> ### Author Response · Authors · 2024-11-24
>
> Dear Rev QhxJ,
> as the discussion period approaches its conclusion, we wanted to ask if you have any additional questions or comments.
>
> In short, we made several key revisions to the paper. First, we added a pipeline diagram to better illustrate the approach and reorganized the experimental results to clearly separate the main findings from the ablation studies. We also expanded the discussion in Appendix C regarding the distillation experiments, providing additional insights into the results. Furthermore, we clarified the distinction between our "time-then-graph" approach and similar methods like TGN, emphasizing our unique use of boosted decision trees for temporal components. Lastly, we addressed minor typographical errors, particularly in tables, and corrected them in the revision.

---

> > ### Comment · Reviewer_QhxJ · 2024-11-26
> >
> > Thank you for your response. My concerns regarding the method illustration and experiment demonstration have been resolved. However, the explanations on how the TREeLGNN shows advantages are still a bit vague for me. Although I agree that it is not necessary to fully discuss these in one work, I still think it impact the contribution. Therefore, I tend to maintain my current score.

---

### Author Response · Authors · 2024-11-24

We thank all reviewers for their valuable feedback and are delighted by the consistent recognition of our method's runtime and inference efficiency, with Reviewer bySj and Reviewer EvAv emphasizing its significant speedup and practical relevance. We also appreciate that Reviewer qwSi and Reviewer EvAv found our paper clear and intuitive, and thank Reviewer QhxJ for highlighting the flexibility and extendability of our framework. Finally, we are encouraged by Reviewer qwSi and Reviewer bySj's acknowledgment of our method’s strong predictive performance and efficiency on relational database tasks.

Thanks to the insightful feedback from the reviewers, we have made the following revisions to the manuscript, which are highlighted in red in the revised version:
- **Dataset Expansion and Scope:** Added the "Amazon" dataset, expanding evaluation to 5 datasets and 15 tasks; clarified dataset limitations and pipeline flexibility.
 - **Performance and Efficiency:** Explained speedup due to pretrained LightGBM and reduced graph complexity.
 - **Distillation Process and Experiments:** Expanded explanation of distillation and its effectiveness, with additional results in Appendix C.
 - **Baseline Comparison:** Demonstrated TREeLGNN’s consistent improvement over LightGBM in most tasks, and clarified RDL comparison.
 - **Methodology Clarifications:** Explained TREeLGNN’s use of pretrained LightGBM and clarified batch processing differences with RDL.
 - **Reorganization and Updates:** Added pipeline diagram, reorganized experimental section, and included missing appendices.
 - **Time-Then-Graph Clarification:** Clarified the role of LIGHTGBM in the "time-then-graph" framework.
 - **Corrections:** Fixed typographical errors and revised misleading title about speedup.

---

### Meta-Review · Area_Chair_pznC · 2024-12-20

**Metareview:**

This paper combines pretrained LightGBM models with static GNNs to improve efficiency in relational deep learning but falls short on several fronts. The approach lacks novelty, with similar methods already explored, and its experimental evaluation is limited to a small subset of tasks and datasets, reducing generalizability. Claims of significant runtime efficiency are overstated and not well-supported by fair comparisons. Additionally, the paper’s focus on temporal relational databases limits its broader applicability. Presentation issues, including unclear distinctions from baselines and insufficient explanation of figures, further weaken the work. Overall, the contribution is not sufficient for acceptance at this stage.

**Additional Comments On Reviewer Discussion:**

During the rebuttal period, reviewers raised the following points:

1. Novelty: Concerns about the lack of innovation compared to existing methods, especially the integration of tree-based models with GNNs.
Authors' Response: Clarified differences with prior work, emphasizing the use of pretrained LightGBM for temporal modeling and reduced graph complexity. Added discussions to distinguish their approach.
Assessment: The response provided some clarification but did not convincingly demonstrate substantial novelty.

2. Experimental Scope: Limited datasets and tasks, with missing baselines like XGBoost and static GNNs.
Authors' Response: Added the "Amazon" dataset and tasks, increasing coverage to 15 tasks. Justified the choice of baselines based on available methods.
Assessment: The additions improved coverage but did not sufficiently address the lack of baseline diversity.

3. Efficiency Claims: Overstated "100x speedup" and unclear comparisons of runtime with preprocessing times excluded.
Authors' Response: Revised the title to avoid misleading claims and added runtime comparisons including preprocessing times.
Assessment: The revision clarified efficiency metrics but raised doubts about fairness in runtime comparisons.

Final Decision: While the authors made commendable efforts to address the feedback, the responses were insufficient to overcome concerns about novelty, experimental scope, and broad applicability. Thus, the paper remains below the acceptance threshold.

---

### Decision · Program_Chairs · 2025-01-22

Reject